# Chameleon: Plug-and-Play Compositional Reasoning with Large Language Models

**Pan Lu**[1], **Baolin Peng**[2], **Hao Cheng**[2], **Michel Galley**[2]
**Kai-Wei Chang**[1], **Ying Nian Wu**[1], **Song-Chun Zhu**[1], **Jianfeng Gao**[2]
[1]University of California, Los Angeles    [2]Microsoft Research, Redmond
https://chameleon-llm.github.io

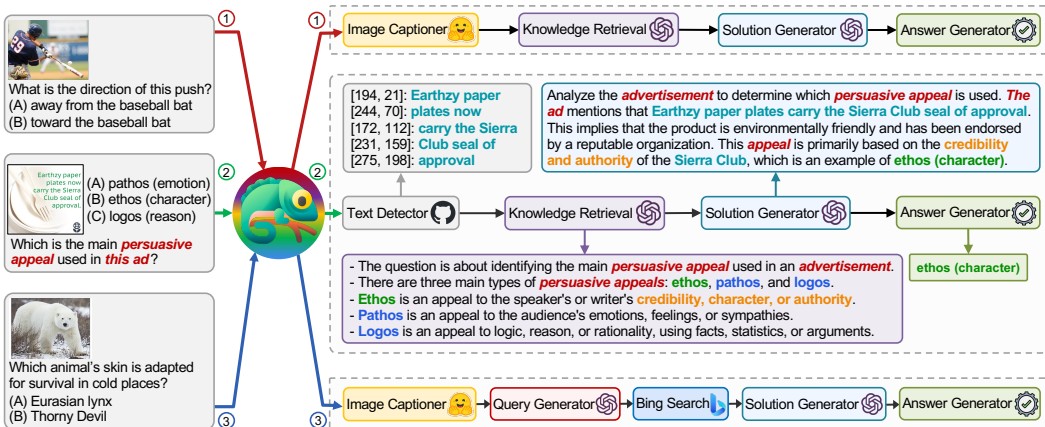

Figure 1: Examples from our **Chameleon** approach with GPT-4 on ScienceQA [32], a multi-modal question answering benchmark in scientific domains. **Chameleon** is adaptive to different queries by synthesizing programs to compose various tools and executing them sequentially to get final answers.

## Abstract

Large language models (LLMs) have achieved remarkable progress in solving various natural language processing tasks due to emergent reasoning abilities. However, LLMs have inherent limitations as they are incapable of accessing up-to-date information (stored on the Web or in task-specific knowledge bases), using external tools, and performing precise mathematical and logical reasoning. In this paper, we present **Chameleon**, an AI system that mitigates these limitations by augmenting LLMs with *plug-and-play* modules for compositional reasoning. **Chameleon** synthesizes programs by composing various tools (e.g., LLMs, off-the-shelf vision models, web search engines, Python functions, and heuristic-based modules) for accomplishing complex reasoning tasks. At the heart of **Chameleon** is an LLM-based planner that assembles a sequence of tools to execute to generate the final response. We showcase the effectiveness of **Chameleon** on two multi-modal knowledge-intensive reasoning tasks: ScienceQA and TabMWP. **Chameleon**, powered by GPT-4, achieves an 86.54% overall accuracy on ScienceQA, improving the best published few-shot result by 11.37%. On TabMWP, GPT-4-powered **Chameleon** improves the accuracy by 17.0%, lifting the state of the art to 98.78%. Our analysis also shows that the GPT-4-powered planner exhibits more consistent and rational tool selection via inferring potential constraints from instructions, compared to a ChatGPT-powered planner.

This title draws inspiration from the *chameleon*'s ability to adapt and blend into its surroundings, which parallels the adaptability and versatility of large language models in compositional reasoning tasks with external tools.

37th Conference on Neural Information Processing Systems (NeurIPS 2023).

# 1 Introduction

Remarkable progress has been observed in recent large language models (LLMs) for various natural language processing tasks, with prominent examples such as GPT-3 [4], PaLM [8], LLaMA [64], ChatGPT [41], and the recently developed GPT-4 [42]. LLMs have demonstrated emergent abilities, including in-context learning and chain-of-thought (CoT) reasoning [56]. These models are capable of solving diverse tasks in a zero-shot fashion [25] or with the aid of a few examples [57], and they show great potential in planning and decision-making akin to human beings [17, 16]. Despite these capabilities, LLMs face inherent limitations, such as an inability to access up-to-date information [26], perform precise mathematical reasoning [44, 35], or utilize specialized models [49]. Therefore, enhancing current LLMs with the capability to automatically *compose* external tools for real-world task solving is critical to address these drawbacks.

Consider the example ② in Figure 1: *Which is the main persuasive appeal used in this ad?*. To answer this question, one needs to: 1) infer that there is an ad image containing text context and call a text decoder to understand the semantics; 2) retrieve background knowledge about *persuasive appeals* and the differences among three persuasive appeals; 3) generate a solution based on the input query and intermediate results from previous steps; and 4) finally produce the answer in a task-specific format. On the other hand, when answering *Which animal's skin is adapted for survival in cold places* (③), one might need to call modules such as an image captioner to decipher image information and a web search engine to retrieve domain knowledge to understand scientific terminologies. However, current tool-augmented LLMs still face challenges when addressing these real-world queries across various scenarios. Most existing approaches are either limited to a small number of tools [39, 6, 55, 18, 43, 49] or relying on domain-specific tools [40, 60, 13, 59, 52], and thus are not easy to generalize to queries of new domains (see sections 2 and A.1 for further discussion). In this work, we study how to enable LLMs to synthesize programs to capture the logic of composing heterogeneous tools.

To address the challenges of existing work, we introduce **Chameleon**, a *plug-and-play compositional* reasoning framework that leverages LLMs to synthesize programs and compose various tools for a wide range of tasks. Unlike existing tool-augmented LLMs [49, 40, 60, 13, 59, 52], **Chameleon** uses a richer set of tools, including LLMs, off-the-shelf vision models, web search engines, Python functions, and heuristics-based modules. Moreover, **Chameleon** leverages the in-context learning capabilities of LLMs and builds on an LLM as a natural language planner, without requiring any training or carefully curated rules. Prompted by tool descriptions and usage examples, the planner infers a program composed of a sequence of tools to execute in order to generate the final response for a user query. Instead of generating programs in domain-specific languages [40, 52, 13], **Chameleon** generates natural-language-like (NL) programs (e.g., [Text_Detector, Knowledge_Retrieval, Solution_Generator, Answer_Generator] for the second query in Figure 1). The NL-like programs are easy to understand and debug by users with limited programming experience, and easily extendable to new modules. During each module's execution, the module processes the query and cached context, returns a result determined by the module itself, and updates the query and context for subsequent execution. Composing modules as a sequential program allows subsequent modules to leverage prior cached context and updated queries.

We showcase the adaptability and effectiveness of **Chameleon** on two tasks: ScienceQA [32] and TabMWP [33]. ScienceQA is a multi-modal question answering benchmark spanning multiple context formats and various scientific topics, while TabMWP is a mathematical benchmark involving diverse tabular contexts. These two benchmarks serve as a good testbed to evaluate **Chameleon**'s ability to coordinate diverse tools across different types and domains. Notably, **Chameleon** with GPT-4 achieves an 86.54% accuracy on ScienceQA, significantly improving upon the best published few-shot model by 11.37%. On TabMWP, using GPT-4 as the underlying LLM, **Chameleon** achieves an improvement of 7.97% over chain-of-thought (CoT) prompted GPT-4 [57] and a 17.0% increase over the best-published model [6], lifting the state of the art to 98.78%. Further studies suggest that using GPT-4 as a planner exhibits more consistent and rational tool selection and is able to infer potential constraints given the instructions, compared to other LLMs like ChatGPT.

Our contributions are as follows: (1) We develop a plug-and-play compositional reasoning framework, **Chameleon**, that effectively composes external tools to address inherent limitations of LLMs and tackle a broad range of reasoning tasks. (2) Relying on an LLM as a natural language planner to generate programs, **Chameleon** successfully integrates various tools, including LLMs, off-the-shelf vision models, web search engines, Python functions, and rule-based modules, to build a versatile and

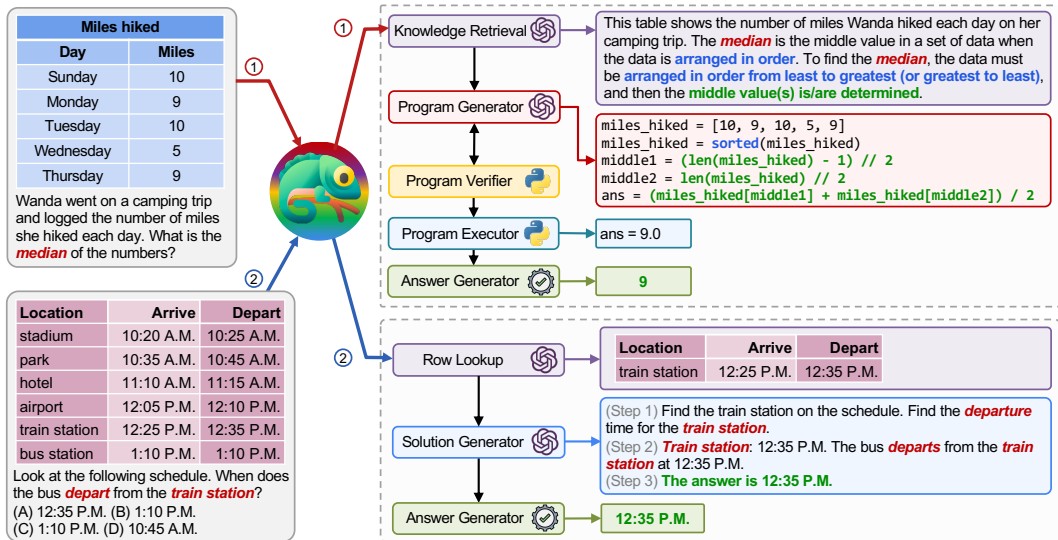

Figure 2: Two examples from our **Chameleon** approach with GPT-4 on TabMWP [33], a mathematical reasoning benchmark with tabular contexts. **Chameleon** demonstrates flexibility and efficiency in adapting to different queries that require various reasoning abilities.

adaptable AI system capable of answering real-world queries. (3) We demonstrate **Chameleon**'s effectiveness on two challenging benchmarks, significantly surpassing the state of the art.

## 2 Related Work

**Compositional Reasoning**  Neural modular and compositional approaches have been explored to automatically perform desired sub-task decomposition, enhancing interpretability and adaptability across various reasoning tasks. Early work [2, 3] posits that complex reasoning tasks are fundamentally compositional and proposes neural module networks (NMN) to decompose them into subtasks. However, these methods rely on brittle off-the-shelf parsers and are limited by module configurations. Some later work [19, 15, 14, 21], takes a step further by predicting instance-specific network layouts in an end-to-end manner, without relying on parsers, using reinforcement learning [58] and weak supervised learning. In visual reasoning, models comprising a program generator and an execution engine have been proposed to combine deep representation learning and symbolic program execution [19, 61]. In the domain of mathematical reasoning, an interpretable solver has been developed to incorporate theorem knowledge as conditional rules and perform symbolic reasoning step by step [31]. Our work takes inspiration from neural module networks, yet it offers several distinct advantages. First, **Chameleon** does not require expensive supervision of task-specific programs for modeling training. Instead, it generates sequential programs, consisting of modules, that are easy to generalize to various domains and tasks, allowing the extension to new modules in a plug-and-play manner. Second, **Chameleon** does not require any training, but uses the in-context learning capabilities of LLMs to generate programs prompted by natural language instruction and demonstrations.

**Tool-Augmented Language Models**  In recent years, the development of large language models (LLMs) [48, 8, 9, 53, 4, 41, 42] has made tremendous progress and has stimulated research in prompt learning [57, 33, 22] and instruction learning [53, 64, 46, 11]. Despite the impressive performance of LLMs, they suffer from inherent limitations, such as the inability to access up-to-date information [26], utilize external tools [49], or perform precise mathematical reasoning [44, 35]. Recent benchmarks, such as ScienceQA and TabMWP [32, 33, 7, 54, 51, 30], have emerged to evaluate the capability of LLMs to tackle intricate reasoning challenges, especially those emphasizing the use of external tools. Concurrently, there has been a growing interest in harnessing external tools and modular approaches to augment LLMs. These augmented LLMs can access real-time information aided by web search engines [40] and leverage domain-specific knowledge from external resources [62]. Some work leverages the Python interpreter to generate complex programs to employ powerful computational resources, and execute logical reasoning tasks more effectively [55, 10, 6, 39, 18, 43, 36]. For example, Toolformer [49] constructs tool-use augmented data to train language models to select five

| Model | Tool Use | | | | | | Skill Dimension | | | | | Inference & Extension | | |
|---|---|---|---|---|---|---|---|---|---|---|---|---|---|---|
| | Size | 🌐 | 🤗 | ⚙ | b | 🐍 | Image | Web | Know. | Math | Table | Composition | Planning | Plug-n-Play |
| CoT [57] | 1 | ✓ | ✗ | ✗ | ✗ | ✗ | ✗ | ✗ | ✗ | ✓ | ✗ | ✗ | ✗ | ✗ |
| Lila [39] | 1 | ✓ | ✗ | ✗ | ✗ | ✓ | ✗ | ✗ | ✗ | ✓ | ✗ | ✗ | ✗ | ✗ |
| PoT [6] | 2 | ✓ | ✗ | ✗ | ✗ | ✓ | ✗ | ✗ | ✗ | ✓ | ✗ | ✗ | ✗ | ✗ |
| Code4Struct [55] | 1 | ✓ | ✗ | ✗ | ✗ | ✓ | ✗ | ✗ | ✗ | ✗ | ✗ | ✗ | ✗ | ✗ |
| PAL [10] | 2 | ✓ | ✗ | ✗ | ✗ | ✓ | ✗ | ✗ | ✗ | ✓ | ✗ | ✗ | ✗ | ✗ |
| MathPrompter [18] | 2 | ✓ | ✗ | ✗ | ✗ | ✓ | ✗ | ✗ | ✗ | ✓ | ✗ | ✗ | ✗ | ✗ |
| ART [43] | 4 | ✓ | ✗ | ✗ | ✓ | ✓ | ✗ | ✓ | ✗ | ✓ | ✗ | ✓ | ✗ | ✓ |
| Toolformer [49] | 5 | ✗ | ✗ | ✗ | ✓ | ✗ | ✗ | ✓ | ✗ | ✗ | ✗ | ✗ | natural lang. | ✗ |
| WebGPT [40] | 10 | ✓ | ✗ | ✗ | ✓ | ✗ | ✗ | ✓ | ✗ | ✗ | ✗ | ✓ | program | ✗ |
| MM-ReAct [60] | >10 | ✓ | ✗ | ✗ | ✓ | ✗ | ✓ | ✓ | ✓ | ✓ | ✓ | ✓ | word match | ✓ |
| Visual ChatGPT [59] | >10 | ✓ | - | - | ✗ | ✗ | ✓ | ✗ | ✗ | ✗ | ✗ | ✓ | natural lang. | ✓ |
| ViperGPT [52] | >10 | ✓ | - | - | ✗ | ✗ | ✓ | ✗ | ✓ | ✓ | ✗ | ✓ | program | ✓ |
| VisProg [13] | >10 | ✓ | - | - | ✗ | ✓ | ✓ | ✗ | ✗ | ✗ | ✗ | ✓ | program | ✓ |
| HuggingGPT [50] | >10 | ✓ | ✓ | ✗ | ✗ | ✗ | ✓ | ✗ | - | ✗ | - | ✓ | natural lang. | ✓ |
| **Chameleon (ours)** | >10 | ✓ | ✓ | ✓ | ✓ | ✓ | ✓ | ✓ | ✓ | ✓ | ✓ | ✓ | natural lang. | ✓ |

Table 1: A comparison of work that augments large language models with tool usage. We report the tool size and tool types, including OpenAI (🌐), Hugging Face (🤗), Github (⚙), Web search (b), and code (🐍). We compare the skills each method possesses, such as image understanding, browser search, knowledge retrieval, mathematical reasoning, and table understanding. Some models can compose various tools, propose a planner to infer the relevant tools for execution, or are inherently extendable to new tools. The label "-" refers to uncertain information in the literature.

tools. In the realm of visual tools, various approaches have been proposed to enhance the capabilities of large language models in handling visual tasks [60, 59, 52, 13, 50], augmented with Hugging Face models [50], Azure models [60], visual foundation models [59].

We compare **Chameleon** with other tool-augmented language models in Table 1. Many of these approaches are either constrained to a small set of tools or limited to task-specific tools, which reduces their capabilities across various skill dimensions and hampers their generalizability to new tasks. A recent line of work relies on large amounts of supervision [49, 26] and focuses on generating commands [40] and programs [52, 13] to infer the choice of tools. However, this approach needs to carefully tailored prompts to specific tasks and particular tools, and is neither flexible nor adaptive. In contrast, **Chameleon** instructs LLMs with natural language instructions that simply describe the roles of each module and provide a few calling examples, eliminating the need for additional training or tool-specific prompts when learning to compose different tools. More importantly, **Chameleon** offers users flexibility in terms of tool types and sources, updating the underlying LLMs, adding new tools, and adapting to new tasks. Our work shares the same spirit of AutoGPT [47], an autonomous GPT-4 agent with the artificial general intelligence (AGI) ambition to incorporate numerous tools to achieve user-defined goals. While AutoGPT is still under development, our work is the first to instantiate the idea and verify its effectiveness on well-studied benchmarks.

## 3 General Framework: **Chameleon**

To address the limitations of current LLMs in utilizing diverse tools, we propose **Chameleon**, a novel *plug-and-play compositional* reasoning framework, synthesizing the composition of various tools to accommodate a wide range of problems. **Chameleon** is comprised of a *module inventory* that defines different types of tools and an LLM-based *planner*, whose purpose is to decompose the original problem into sub-tasks that can be effectively solved by task-specific tools. Unlike existing tool-augmented LLM approaches [49, 13, 59, 50], our module inventory features multiple tool types as illustrated in Table 2, enabling **Chameleon** to exhibit various reasoning abilities, including image understanding, knowledge retrieval, web search, complex mathematical reasoning, and table understanding. Instead of generating domain-specific programs [40, 13, 52], **Chameleon** employs an LLM-based planner to create natural-language-like programs that follow natural language instructions, which is less error-prone, easily expandable to new modules, and user-friendly.

We formalize our planner as follows: given the input query $x_0$, the module inventory $\mathcal{M}$, and constraints $\mathcal{G}$, the natural language planner $\mathcal{P}$ selects a set of modules that can be executed sequentially to answer the query via generating a program in a natural-language-like format. The module inventory $\mathcal{M}$ consists of a set of pre-built modules: $\{M_i\}$, each corresponding to a tool of various types (Table 2). $\mathcal{G}$ are the constraints for the plan generation, for example, the concurrent relations and sequence

orders of modules. In our work, the planner $\mathcal{P}$ is an LLM prompted to generate a sequence of module names in a few-shot setup. The planner is prompted in natural language with a planning task instruction $\mathcal{I}$, the descriptions of modules in $\mathcal{M}$ with corresponding constraints $\mathcal{G}$, as well as a few demonstration examples $\mathcal{D}$. A $T$-length plan sampled from $\mathcal{P}$ can be denoted as $p = M^1, \ldots, M^T$, where $M^t$ represents an the $t$-th element in the generated plan and $M^t \in \mathcal{M}$. Formally, given an input query (problem statement) $x_0$, a plan $p$ is generated as follows:

$$p \leftarrow \mathcal{P}(x_0; \mathcal{I}, \mathcal{M}, \mathcal{G}, \mathcal{D}). \tag{1}$$

Given the generated plan, the corresponding modules for each step are then executed sequentially. The plan is a natural-language program where each module is bound simply via string matching. When evaluating the module $M^t$ at time step $t$, the output of the execution $y^t$ is calculated by:

$$y^t \leftarrow M^t(x^{t-1}; c^{t-1}), \tag{2}$$

where $x^{t-1}$ is the input for the current module $M^t$, and $c^{t-1}$ is the cached information (e.g., image semantics, retrieved knowledge, generated programs) resulting from the execution history of modules. Both the problem input $x^t$ and cache $c^t$ for the next module $M^{t+1}$ are updated, respectively, by:

$$x^t \leftarrow \texttt{update\_input}(x^{t-1}, y^t), \tag{3}$$

$$c^t \leftarrow \texttt{update\_cache}(c^{t-1}, y^t). \tag{4}$$

The $\texttt{update\_input}$ and $\texttt{update\_cache}$ functions are hand-designed for each $M_i$. Specifically, $\texttt{update\_input}$ is applied to elements in the input query, including the question, table context, and image. These elements are updated after module execution. $\texttt{update\_cache}$ corresponds to the generation of new information, such as a description for the input image or retrieved knowledge from external resources. Finally, the response $r$ to the query is generated by the last module $M^T$:

$$r = y^T \leftarrow M^T(x^{T-1}; c^{T-1}). \tag{5}$$

## 4 Applications of Chameleon

We demonstrate the applications of **Chameleon** on two challenging tasks: ScienceQA [32] (section 4.2) and TabMWP [33] (section 4.3), using the module inventory introduced in section 4.1. Further experimental details can be found in appendix A.2.

### 4.1 Module Inventory

To accommodate various reasoning capabilities over a diverse range of queries, our system utilizes a rich module inventory of various external tools. We provide a high-level overview of this inventory here, with detailed implementations in specific experiments. The complete module inventory, $\mathcal{M}$, is presented in Table 2. Each tool within the inventory is defined as follows:

**Knowledge Retrieval**: This module retrieves additional background knowledge crucial for tackling complex problems. It is especially beneficial for specialized domains like science and mathe-

| Tool Types | Tools |
|---|---|
| OpenAI | Knowledge Retrieval, Query Generator, Row Lookup, Column Lookup, Table Verbalizer, Program Generator, Solution Generator |
| Hugging Face | Image Captioner |
| Github | Text Detector |
| Web Search | Bing Search |
| Python | Program Verifier, Program Executor |
| Rule-based | Answer Generator |

Table 2: Different tools in our module inventory.

matics, providing context for the task. For example, if a query is about a tax form table, this module could generate knowledge about tax procedures, offering valuable context.

**Bing Search**: Like "Knowledge Retrieval", the "Bing Search" module aims to provide wide-ranging task-relevant knowledge. In contrast, it excels when broader or up-to-date information from multiple sources is required. Using the search engine API, this module returns relevant search results based on the input query, which are then parsed and used by subsequent modules to gather richer context information from diverse sources, enhancing problem-solving effectiveness.

**Query Generator**: Since the original problem typically lacks a tailored query for retrieving task-relevant information, this module creates search engine queries based on the problem, which are then

used by the "Bing Search" module. Mostly, it is a good strategy to use the "Query Generator" module before the "Bing Search". Coupled with the search engine tool, generating more targeted queries generally facilitates both the recall and precision of retrieved information.

**Image Captioner**: Designed to generate captions for images, this module provides crucial supplementary context for queries. It is particularly valuable when understanding an image semantically, like identifying objects and interactions in a scene. Using pre-trained models, it translates visual data into language, facilitating effective comprehension and reasoning about image content.

**Text Detector**: This module is designed to identify text within a given image. Typically, the "Text Detector" is employed when a question requires the extraction of textual information from images containing diagrams, charts, tables, maps, or other visual elements. By effectively detecting text in various formats, this module aids in the analysis and understanding of image-based content.

**Row Lookup**: This module is crucial when queries involve tabular context, as locating relevant cells is often required. Large tables can distract the system, so "Row Lookup" simplifies the table by retaining only the rows relevant to the query. If all rows are pertinent, it returns the original table.

**Column Lookup**: Like the "Row Lookup" module, "Column Lookup" addresses questions involving tabular context by focusing on relevant columns. It simplifies the table by retaining only pertinent columns, or returns the original table if all columns are relevant.

**Table Verbalizer**: Converting structured tables into text is likely to enhance the comprehension of tabular information by various downstream modules as shown by [37] for open-domain question answering, making this module a vital part of our system. It translates tables into easily understandable descriptions for modules like "Program Generator" and "Solution Generator", particularly useful for small, domain-specific tables like stem-and-leaf plots or function tables.

**Program Generator**: Program-aided approaches are shown to enhance the logical and mathematical reasoning abilities of LLMs [55, 10, 6, 39, 18, 43]. The "Program Generator" generates Python programs to solve queries effectively, which is particularly beneficial for queries requiring complex computations or intricate logical operations, such as "if-else" statements.

**Program Verifier**: Recent studies highlight the importance of verification to reduce hallucination [45, 38]. Hence, "Program Verifier" ensures the validity and error-free nature of programs generated by "Program Generator". It checks for syntax and logical errors, and potential execution issues, enhancing the reliability and accuracy of the solutions.

**Program Executor**: This module executes the program generated by "Program Generator" and produces the result, bridging the gap between program generation and final solution derivation.

**Solution Generator**: This module generates a detailed solution to the input query using all the cached information. Employing a chain-of-thought prompting approach [57], it ensures coherent and well-structured responses. The planner can directly employ this module instead of other functional modules if it can solve the query independently, especially for simpler ones.

**Answer Generator**: This task-specific module uses a rule-based approach to extract and normalize answers from the results of the "Program Executor" or "Solution Generator". Unlike the Solution Generator" that provides detailed multi-step solutions, "Answer Generator" serves as the final module in the pipeline, providing concise and task-specific answers.

## 4.2  Science Question Answering

Science Question Answering (ScienceQA [32]) is a diverse benchmark for multi-modal question answering over a range of scientific topics and contexts. As examples illustrated in Figure 1, answering these questions requires various tools and skills like image captioning, text detection, knowledge retrieval, online resource search, and multi-clue visual reasoning. When generating programs for using tools, we limit the search space to the relevant inventory subset (Table 6 in the appendix). Programs are deemed invalid and default to a "Solution Generator" and "Answer Generator" sequence if these are not the final two elements, following the chain-of-thought prompting baseline [57]. See Table 8 in the appendix for the constructed natural language planner prompt. The prompts for LLM-based modules like "Knowledge Retrieval", "Query Generator", and "Solution Generator" are shown in Table 10, 11, and 12, respectively, in the appendix.

| Model | #Tuned Params | ALL | NAT | SOC | LAN | TXT | IMG | NO | G1-6 | G7-12 |
|---|---|---|---|---|---|---|---|---|---|---|
| *Heuristic baselines* | | | | | | | | | | |
| Random Choice [32] | - | 39.83 | 40.28 | 46.13 | 29.25 | 47.45 | 40.08 | 33.66 | 39.35 | 40.67 |
| Human [32] | - | 88.40 | 90.23 | 84.97 | 87.48 | 89.60 | 87.50 | 88.10 | 91.59 | 82.42 |
| *Fine-tuned models* | | | | | | | | | | |
| MCAN [63] | 95M | 54.54 | 56.08 | 46.23 | 58.09 | 59.43 | 51.17 | 55.40 | 51.65 | 59.72 |
| Top-Down [1] | 70M | 59.02 | 59.50 | 54.33 | 61.82 | 62.90 | 54.88 | 59.79 | 57.27 | 62.16 |
| BAN [23] | 112M | 59.37 | 60.88 | 46.57 | 66.64 | 62.61 | 52.60 | 65.51 | 56.83 | 63.94 |
| DFAF [12] | 74M | 60.72 | 64.03 | 48.82 | 63.55 | 65.88 | 54.49 | 64.11 | 57.12 | 67.17 |
| ViLT [24] | 113M | 61.14 | 60.48 | 63.89 | 60.27 | 63.20 | 61.38 | 57.00 | 60.72 | 61.90 |
| Patch-TRM [34] | 90M | 61.42 | 65.19 | 46.79 | 65.55 | 66.96 | 55.28 | 64.95 | 58.04 | 67.50 |
| VisualBERT [27, 28] | 111M | 61.87 | 59.33 | 69.18 | 61.18 | 62.71 | 62.17 | 58.54 | 62.96 | 59.92 |
| UnifiedQA [20] | 223M | 70.12 | 68.16 | 69.18 | 74.91 | 63.78 | 61.38 | 77.84 | 72.98 | 65.00 |
| UnifiedQA CoT [32] | 223M | 74.11 | 71.00 | 76.04 | 78.91 | 66.42 | 66.53 | 81.81 | 77.06 | 68.82 |
| MM-COT$_T$ [65] | 223M | 70.53 | 71.09 | 70.75 | 69.18 | 71.16 | 65.84 | 71.57 | 71.00 | 69.68 |
| MM-COT [65] | 223M | 84.91 | 87.52 | 77.17 | 85.82 | 87.88 | 82.90 | 86.83 | 84.65 | 85.37 |
| MM-COT$_{Large}$ [65] | 738M | 91.68 | 95.91 | 82.00 | 90.82 | 95.26 | 88.80 | 92.89 | 92.44 | 90.31 |
| LLaMA-Adapter$_T$ [64] | 1.2M | 78.31 | 79.00 | 73.79 | 80.55 | 78.30 | 70.35 | 83.14 | 79.77 | 75.68 |
| LLaMA-Adapter [64] | 1.8M | 85.19 | 84.37 | 88.30 | 84.36 | 83.72 | 80.32 | 86.90 | 85.83 | 84.05 |
| *Few-shot GPT-3* | | | | | | | | | | |
| GPT-3 [4] | 0M | 74.04 | 75.04 | 66.59 | 78.00 | 74.24 | 65.74 | 79.58 | 76.36 | 69.87 |
| GPT-3 CoT [32] | 0M | 75.17 | 75.44 | 70.87 | 78.09 | 74.68 | 67.43 | 79.93 | 78.23 | 69.68 |
| Published results (Above) ▲ | | | | | | | | | | |
| *Few-shot ChatGPT* | | | | | | | | | | |
| ChatGPT CoT | 0M | 78.31 | 78.82 | 70.98 | 83.18 | 77.37 | 67.92 | 86.13 | 80.72 | 74.03 |
| **Chameleon** (ChatGPT) | 0M | 79.93 | 81.62 | 70.64 | 84.00 | 79.77 | 70.80 | 86.62 | 81.86 | 76.53 |
| *Few-shot GPT-4* | | | | | | | | | | |
| GPT-4 CoT | 0M | 83.99 | 85.48 | 72.44 | 90.27 | 82.65 | 71.49 | 92.89 | 86.66 | 79.04 |
| **Chameleon** (GPT-4) | 0M | **86.54** | **89.83** | **74.13** | **89.82** | **88.27** | **77.64** | **92.13** | **88.03** | **83.72** |

Table 3: **QA accuracy (%) on the test set of ScienceQA** [32]. We report the number of tuned parameters for this task and the overall accuracy, along with accuracy scores for different question types, including natural, social, and language sciences, text, image, and no context, as well as grades 1-6 and 7-12. The highest scores among models in each section and overall are highlighted in blue and red, respectively, and the results of our best model are marked in **bold**.

## 4.3 Tabular Mathematical Reasoning

TabMWP [33] is a mathematical reasoning task involving diverse tabular contexts like schedules, prices, tax forms, plots, and function relations (Figure 2). It requires AI systems to understand various table formats and perform precise numerical or symbolic computations. Like ScienceQA, we constrain the program search space to focus on two tool types: 1) those helping LLMs better digest tabular information (e.g., "Row Lookup", "Column Lookup", and "Table Verbalizer") and 2) those performing faithful symbolic computations (e.g., "Program Generator", "Program Verifier", and "Program Executor") as listed in Table 6. The generated programs must meet certain constraints, such as including "Answer Generator" and placing "Program Generator" prior to both "Program Verifier" and "Program Executor". Non-compliant programs default to a sequence of "Program Generator", "Program Verifier", "Program Executor", and "Answer Generator", aligning with the program-of-thought prompting baseline [6] with added verification.

## 5 Experiments

We assess **Chameleon**'s effectiveness and adaptability on two complex reasoning tasks, ScienceQA [32] and TabMWP [33]. See experimental details in appendix A.2.

### 5.1 Experimental Results

**ScienceQA.** Table 3 presents the results of existing baselines and our approach **Chameleon**, with key results highlighted in Figure 3 (a). Employing ChatGPT [41] as the base LLM, **Chameleon**

| Model | #Tuned Params | ALL | FREE | MC | INT | DEC | EXTR | BOOL | OTH | G1-6 | G7-8 |
|---|---|---|---|---|---|---|---|---|---|---|---|
| *Heuristic baselines* | | | | | | | | | | | |
| Heuristic guess | - | 15.29 | 6.71 | 39.81 | 8.37 | 0.26 | 30.80 | 51.22 | 26.67 | 17.55 | 12.27 |
| Human performance | - | 90.22 | 84.61 | 93.32 | 84.95 | 83.29 | 97.18 | 88.69 | 96.20 | 94.27 | 81.28 |
| *Fine-tuned models* | | | | | | | | | | | |
| UnifiedQA[SMALL] [20] | 41M | 29.79 | 22.27 | 51.31 | 27.27 | 2.83 | 52.28 | 48.11 | 69.52 | 35.85 | 21.71 |
| UnifiedQA[BASE] [20] | 223M | 43.52 | 34.02 | 70.68 | 40.74 | 7.90 | 84.09 | 55.67 | 73.33 | 53.31 | 30.46 |
| UnifiedQA[LARGE] [20] | 738M | 57.35 | 48.67 | 82.18 | 55.97 | 20.26 | 94.63 | 68.89 | 79.05 | 65.92 | 45.92 |
| TAPEX[BASE] [29] | 139M | 48.27 | 39.59 | 73.09 | 46.85 | 11.33 | 84.19 | 61.33 | 69.52 | 56.70 | 37.02 |
| TAPEX[LARGE] [29] | 406M | 58.52 | 51.00 | 80.02 | 59.92 | 16.31 | 95.34 | 64.00 | 73.33 | 67.11 | 47.07 |
| *Zero-shot GPT-3* | | | | | | | | | | | |
| GPT-3 [4] | 0M | 56.96 | 53.57 | 66.67 | 55.55 | 45.84 | 78.22 | 55.44 | 54.29 | 63.37 | 48.41 |
| GPT-3 CoT [57] | 0M | 57.61 | 54.36 | 66.92 | 55.82 | 48.67 | 78.82 | 55.67 | 51.43 | 63.62 | 49.59 |
| *Few-shot GPT-3* | | | | | | | | | | | |
| GPT-3 [4] | 0M | 57.13 | 54.69 | 64.11 | 58.36 | 40.40 | 75.95 | 52.41 | 53.02 | 63.10 | 49.16 |
| GPT-3 CoT [57] | 0M | 62.92 | 60.76 | 69.09 | 60.04 | 63.58 | 76.49 | 61.19 | 67.30 | 68.62 | 55.31 |
| GPT-3 CoT-PromptPG [33] | 0M | 68.23 | 66.17 | 74.11 | 64.12 | 74.16 | 76.19 | 72.81 | 65.71 | 71.20 | 64.27 |
| Codex* [5] | 0M | 59.4 | - | - | - | - | - | - | - | - | - |
| Codex PoT* [6] | 0M | 73.2 | - | - | - | - | - | - | - | - | - |
| Codex PoT-SC* [6] | 0M | 81.8 | - | - | - | - | - | - | - | - | - |
| Published results (Above) ▲ | | | | | | | | | | | |
| *Few-shot ChatGPT* | | | | | | | | | | | |
| ChatGPT CoT | 0M | 82.03 | 78.43 | 92.32 | 75.38 | 90.30 | 92.30 | 92.89 | 87.62 | 83.06 | 80.66 |
| ChatGPT PoT | 0M | 89.49 | 90.24 | 87.35 | 89.31 | 93.82 | 92.10 | 85.89 | 55.24 | 90.60 | 88.00 |
| **Chameleon (ChatGPT)** | 0M | 93.28 | 93.13 | 93.72 | 92.71 | 94.76 | 91.29 | 98.11 | 78.85 | 93.37 | 93.17 |
| *Few-shot GPT-4* | | | | | | | | | | | |
| GPT-4 CoT | 0M | 90.81 | 88.48 | 97.49 | 86.16 | 97.51 | 96.86 | 99.11 | 89.52 | 92.40 | 88.70 |
| GPT-4 PoT | 0M | 96.93 | 97.40 | 95.58 | 98.48 | 93.22 | 96.25 | 98.00 | 68.57 | 96.97 | 96.87 |
| **Chameleon (GPT-4)** | 0M | **98.78** | **98.95** | **98.29** | **99.34** | 97.42 | **98.58** | **98.56** | **93.33** | **98.95** | **98.54** |

Table 4: **QA accuracy (%) on the test set of TabMWP** [33]. We report the number of tuned parameters for this task and the overall accuracy, and accuracy of different question types, including free-text questions, multi-choice questions, integer answers, decimal answers, extractive answers, Boolean answers, other text answers, grades 1-6, and grades 7-8. * refers to a subset of results.

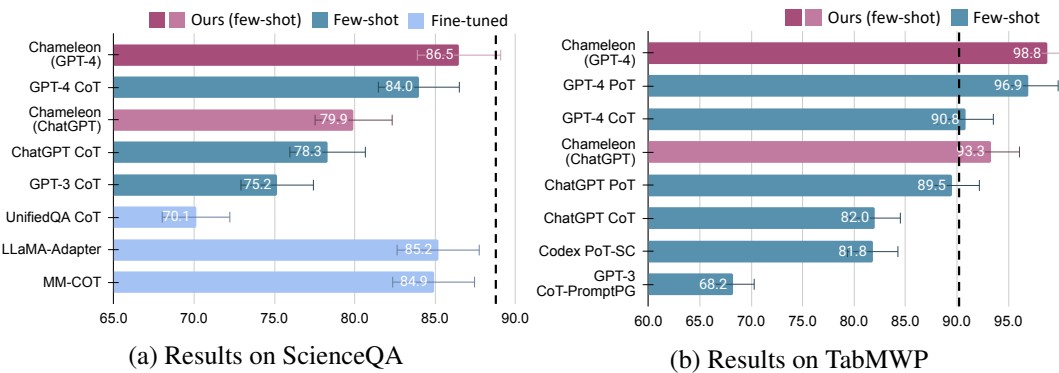

Figure 3: Results of main baselines and **Chameleon**. Dashed lines represent human performance.

achieves a 79.93% accuracy, a 1.62% improvement over Chain-of-Thought (CoT) [57] prompted ChatGPT. Notably, **Chameleon** is a generalized form of CoT, where the generated program is a sequence of "Solution Generator" and "Answer Generator". **Chameleon** benefits from additional tool usage, such as "Knowledge Retrieval", "Bing Search", "Image Captioner", and "Text Detector". When built upon GPT-4 [42], our model attains an accuracy of 86.54%, outperforming GPT-4 CoT [32] by 2.55% and GPT-3 CoT by 11.37%, creating the new state of the art in few-shot settings.

**TabMWP.** Table 4 presents results with key models in Figure 3 (b). Similarly, significant improvements are observed for **Chameleon** over both fine-tuned and few-shot models. It is worth noting that both CoT and Program-of-Thought (PoT) [6] can be viewed as special cases of **Chameleon**. Apart from "Solution Generator" and "Answer Generator", CoT doesn't utilize any tool, while PoT

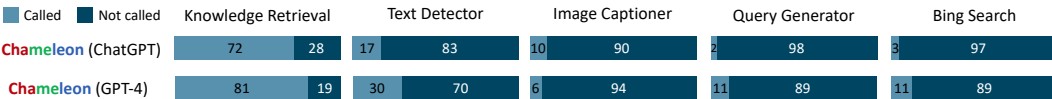

Figure 4: Tools called in the generated programs from **Chameleon** on ScienceQA.

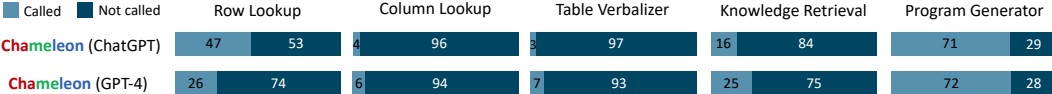

Figure 5: Tools called in the generated programs from **Chameleon** on TabMWP.

only relies on symbolic programming tools like "Program Generator" and "Program Executor". **Chameleon** (ChatGPT) outperforms ChatGPT CoT and ChatGPT PoT by 11.25% and 3.79%, respectively, emphasizing the advantage of our enriched tool set. With GPT-4, **Chameleon** gains an additional 5.50%, reaching a 98.78% accuracy. Notably, **Chameleon** (GPT-4) surpasses Codex PoT-SC [6], the best-published model, by 17.0% and human performance by 8.56%.

## 5.2 Qualitative Analysis

**Tool use planning.** The proportions of key tools called in the programs from **Chameleon** on ScienceQA and TabMWP are visualized in Figure 4 and Figure 5, respectively. Interestingly, Chat-GPT and GPT-4 exhibit different planning behaviors. Generally, ChatGPT has a strong bias toward using or not using certain tools, highly influenced by in-context examples. For instance, ChatGPT calls "Knowledge Retrieval" in 72% of queries but only calls "Bing Search" in 3% of cases on ScienceQA; on TabMWP, ChatGPT heavily relies on "Row Lookup" (47%) but calls "Column Lookup" less frequently (4%). However, GPT-4 acts more *objectively* and *rationally* in tool selection. For example, GPT-4 calls "Knowledge Retrieval" more frequently (81% vs. 72%) and calls "Bing Search" more than ChatGPT (11% vs. 3%) when answering scientific questions on ScienceQA. Impressively, GPT-4 consistently calls "Query Generator" and "Bing Search" simultaneously by observing the tool usage descriptions, while ChatGPT lacks such reasoning capability.

**Ablation study with disabled modules.** We study the accuracy decline of **Chameleon** when key modules in the generated programs are disabled (Table 5), using ChaptGPT as the underlying LLMs and 500 test examples. The results reveal that "Knowledge Retrieval" plays a vital role in both tasks. Domain-specific tools, such as the search engine and vision models for ScienceQA, and program tools for TabMWP, also prove to be important.

| Module | Δ (ScienceQA) | Δ (TabMWP) |
|---|---|---|
| Knowledge Retrieval | -7.8% | -2.2% |
| Bing Search | -7.4% | - |
| Text Detector | -8.4% | - |
| Image Captioner | -6.0% | - |
| Program Generator | - | -7.4% |
| Table Verbalizer | - | -0.2% |

Table 5: Score drop with disabled modules.

**Module transitions.** We visualize the transition graphs of modules for generated programs by **Chameleon** (GPT-4) on ScienceQA and TabMWP in Figure 7 and 8, respectively. The transition probabilities in these graphs are computed from the tool transitions observed on the test sets. These graphs show that the GPT-4 planner is able to make good decisions on how to sequence tools in a few-shot setup. For example, on ScienceQA, **Chameleon** often decides to rely on either "Knowledge Retriever" or "Bing Search", but rarely both. On TabMWP, we observe two main modes: either going through the solution generator module or via the program generator, verifier, and executor.

## 5.3 Case Study

**Visualization examples of ScienceQA.** Examples from **Chameleon** (GPT-4) on ScienceQA are visualized in Figure 1. **Chameleon** (GPT-4) is able to adapt to different input queries by generating programs that compose various tools and executing them sequentially to obtain accurate responses. For instance, to answer the first question (①), *What is the direction of this push?*, the system calls the image captioner model to extract semantic information from the image and employs the knowledge retrieval model to gather background knowledge for multi-modal reasoning. In the second example (②), the natural language planner infers that a text detector tool is needed to understand the context

of the ad. The third query (③; more details provided in Figure 9 in the appendix), *Which animal's skin is adapted for survival in cold places?*, involves scientific terminology related to animal survival. The planner decides to call the Bing search engine to access domain-specific knowledge, benefiting from the numerous online resources.

**Visualization examples of TabMWP.** The adaptability and versatility of **Chameleon** for various queries are also observed on TabMWP, as illustrated in the examples in Figure 2. The first example (①) involves mathematical reasoning on a tax form. **Chameleon** (1) calls the knowledge retrieval model to recall basic knowledge that assists in understanding this domain-specific table, (2) describes the table in a more readable natural language format, and (3) finally relies on program-aided tools to perform precise computations. In the second example (②), the system generates Python code that closely aligns with the background knowledge provided by the knowledge retrieval model. The third example (③) requires the system to locate the cell in a large tabular context given the input query. **Chameleon** calls the row lookup model to help accurately locate the relevant rows and generate the language solution via an LLM model, instead of relying on program-based tools.

**Failure cases and limitations.** Failure examples from **Chameleon** (GPT-4) are illustrated in Tables 19 to 24 in the appendix. Inaccurate responses may arise from the limitations of the current modules or from suboptimal programs generated by the planner. Additionally, the module inventory may lack tools capable of addressing specific abilities. Future directions could involve upgrading the modules and the planner, or expanding the module inventory to support a broader range of capabilities. Further limitations and broader impacts are respectively discussed in sections B and C of the appendix.

## 5.4 Error Analysis

To examine the error sources of the base large language models and understand how our model reduces mistakes from different aspects, we conduct an error analysis, as shown in Figure 6. We select 50 mistake examples from the ChatGPT baseline on ScienceQA as the evaluation set. We count the number of mistake examples and analyze their corresponding mistake type categories for ChatGPT, our **Chameleon** (ChatGPT) approach, and **Chameleon** (GPT-4).

The results show that our **Chameleon** approach can substantially reduce the number of mistakes compared to Chat-GPT. Our model features tools for image captioning and knowledge retrieval, thus the mistakes made by ChatGPT in the category of image understanding are reduced to 10 and 19 from 32 by **Chameleon** (ChatGPT) and **Chameleon** (GPT-4); while the mistakes made by ChatGPT in the category of knowledge understanding are reduced to 6 and 3 from 37 by **Chameleon** (ChatGPT) and **Chameleon** (GPT-4). Benefiting from the sequential execution of tools, the mistakes caused by solution generation are significantly reduced as well. Additionally, we find that the task planning of GPT-4 outperforms ChatGPT by a large margin.

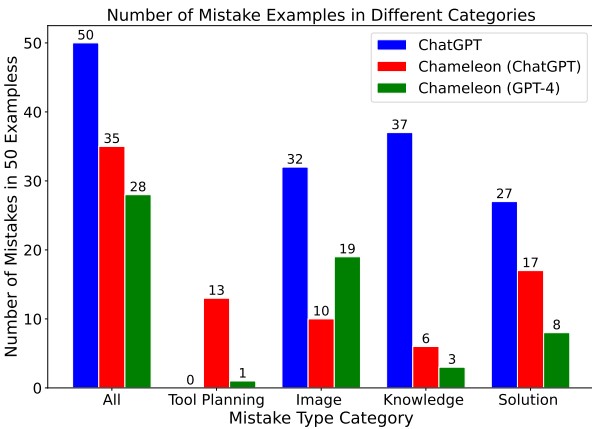

Figure 6: # of mistake examples in different categories on ScienceQA. Image: image captioning, Knowledge: knowledge understanding, Solution: solution generation.

## 6 Conclusion

In conclusion, we introduce a novel *plug-and-play compositional* reasoning framework, **Chameleon**, that addresses the limitations of current large language models by augmenting them with external tools in a plug-and-play manner. Our approach employs a diverse set of tools and demonstrates impressive adaptability and effectiveness on two challenging benchmarks, ScienceQA and TabMWP. By achieving significant improvements in accuracy over existing state-of-the-art models, **Chameleon** showcases its potential for addressing real-world queries across various domains.

# Acknowledgment

We would like to thank Chunyuan Li, Qiuyuan Huang, and other members of the Deep Learning group at Microsoft Research for their valuable discussions. We also thank Fan Yin from University of California, Los Angeles, and Mingyang Sun from University of Electronic Science and Technology of China for their thorough review of our paper and constructive feedback. Pan Lu's research for this work was financially supported by Microsoft during his visit at Microsoft Research, and was also partially supported by the Amazon PhD Fellowship, Bloomberg PhD Fellowship, Qualcomm Innovation Fellowship, and UCLA Dissertation Year Fellowship. Kai-Wei was supported an ONR grant N00014-23-1-2780 and as a Sloan Fellow.

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

# Supplementary Materials for
# Chameleon: Plug-and-Play Compositional Reasoning with Large Language Models

## A  Appendix

### A.1  Current Tool-Augmented LLMs

To address the limitations of LLMs, an active research direction involves augmenting language models with access to external tools and resources, as well as exploring the integration of external tools and plug-and-play modular approaches. For example, aided by web search engines and external knowledge resources, LLMs are able to access real-time information and leverage domain-specific knowledge [40]. To enhance mathematical reasoning abilities, recent work uses LLMs [5] to generate complex programs to exploit powerful computational resources, and execute logical reasoning tasks more effectively [55, 10, 6, 39, 18, 43]. Another line of recent work, such as ViperGPT [52], Visual ChatGPT [59], VisProg [13], and HuggingGPT [50] incorporates a collection of foundation computer vision models to equip LLMs with the abilities to perform visual reasoning tasks.

### A.2  Experimental Details

**Module search space.**  The inventory subsets for ScienceQA and TabMWP are shown in Table 6.

| Tool Types | Tools used on ScienceQA | Tools used on TabMWP |
|---|---|---|
| OpenAI | **Knowledge Retrieval**, Query Generator, **Solution Generator** | **Knowledge Retrieval**, Row Lookup, Column Lookup, Table Verbalizer, Program Generator, **Solution Generator** |
| Hugging Face | Image Captioner | |
| Github | Text Detector | |
| Web Search | Bing Search | |
| Python | | Program Verifier, Program Executor |
| Rule-based | **Answer Generator** | **Answer Generator** |

Table 6: Tools used on ScienceQA and TabMWP, respectively. Reusable tools are marked in **green**.

**Planner implementations.**  We choose the *gpt-3.5-turbo* engine for ChatGPT and the *gpt-4* engine for GPT-4 when constructing the LLM-based planner. The maximum length for generated programs is set to 128, and the temperature is set to 0 for the most deterministic generation. The planner prompts for the ScienceQA and TabMWP are illustrated in Table 8 and Table 9, respectively.

**Module implementations for ScienceQA.**  By default, the LLM-based models use four in-context examples as demonstrations, have a temperature setting of 0, and allow a maximum of 512 tokens for completion. Additional specific implementation details are provided as follows:

- **Knowledge Retrieval**: The prompt consists of 3 demonstration examples and the template is shown in Table 10.

- **Query Generator**: The prompt template is shown in Table 11. The maximum number of tokens for completion is set as 64.

- **Solution Generator**: The prompt consists of 2 demonstration examples and the template is shown in Table 12.

- **Image Captioner**: We use the captioning model[1] to generate textual descriptions for input images. The maximum length of generated captions is set to 16, the number of beams is 4, and the maximum number of output tokens is 512.
- **Text Detector**: This module is based on the github model[2] to extract the text contents with coordinates in the image.
- **Bing Search**: This module calls the Bing Search API[3] and returns the top three responses for the text query.
- **Answer Generator**: This module extracts the answer snippet from the result provided by the "Solution Generator" and selects the most similar option from the given choices.

**Module implementations for TabMWP.** Similar to ScienceQA, the LLM-based modules by default use four in-context examples as demonstrations, have a temperature setting of 0, and allow a maximum of 512 tokens for completion. Additional implementation details are provided as follows:

- **Knowledge Retrieval**: The prompt consists of 5 demonstration examples and the template is shown in Table 13.
- **Row Lookup**: It is enabled only when there are more than three rows and 18 table cells, in order to accelerate inference. The prompt consists of 7 demonstration examples and the template is shown in Table 14. The maximum number of tokens for completion is set as 256.
- **Column Lookup**: Similarly, this module is enabled with two or more columns and 18 or more table cells. The prompt consists of 6 demonstration examples and the template is shown in Table 15. The maximum number of tokens for completion is set as 256.
- **Table Verbalizer**: The prompt consists of 7 demonstration examples and the template is shown in Table 16.
- **Program Generator**: The prompt template is shown in Table 17. The maximum number of tokens for completion is set as 256.
- **Solution Generator**: The prompt consists of 16 demonstration examples and the template is shown in Table 18.
- **Answer Generator**: It is used to normalize answers with two-place precision for questions with numerical answers and select the most similar option for multiple-choice questions.

**Implementations of `update_input` and `update_cache`.** `update_input` is triggered by the execution of specific tools, like 'Row_Lookup', which alter or replace elements in the input to reflect the updated state. Tools such as 'Image_Captioner', 'Text_Detector', 'Knowledge_Retrieval', 'Web_Search', and 'Program_Generation' generate new elements. `update_cache` stores these new elements in the cache, making them accessible for later tools' execution.

### A.3 Experimental Results

**Generated program statistics.** **Chameleon** utilizes the LLM-based natural language planner to generate programs, i.e., sequences of used modules (tools). We report the statistics of the number of unique generated programs and the average length of corresponding tool sequences by **Chameleon** in Table 7. On both ScienceQA and TabMWP, using GPT-4 as the base LLM generates fewer distinct programs, i.e., more consistent programs, than using ChatGPT, even when given the exact same prompt in the planning model. Our results are consistent with the findings in [42], which observes that GPT-4 has a superior capability of understanding long contexts, aligning with human instructions, and performing high-level reasoning compared to other LLMs such as ChatGPT.

## B  Limitations

While **Chameleon** represents a significant stride in exploiting large language models (LLMs) for compositional reasoning in a plug-and-play manner, there are a few areas that could benefit from

---

[1] https://huggingface.co/nlpconnect/vit-gpt2-image-captioning
[2] https://github.com/JaidedAI/EasyOCR
[3] https://www.microsoft.com/bing

| Task | Model | # of different programs | Average program length |
|------|-------|------------------------|------------------------|
| ScienceQA | Chain-of-thought (CoT) | 1 | 2 |
| | **Chameleon** (ChatGPT) | 14 | 3.03 |
| | **Chameleon** (GPT-4) | 11 | 3.40 |
| TabMWP | Chain-of-thought (CoT) | 1 | 2 |
| | Program-of-thought (PoT) | 1 | 3 |
| | **Chameleon** (ChatGPT) | 28 | 4.17 |
| | **Chameleon** (GPT-4) | 19 | 4.09 |

Table 7: The statistics of the number of different generated programs and the average length of generated programs by **Chameleon**, respectively. Chain-of-thought (CoT) prompting and Program-of-thought (PoT) prompting are also compared as they are the special cases of **Chameleon**.

further refinement. One such area is the expansion of its adaptability to a wider variety of tasks and domains, beyond the benchmarks presented. The LLM-based planner, responsible for synthesizing programs and determining the sequence of tools, introduces an innovative approach, yet it also raises intriguing research questions about optimizing the process for tool selection and sequence. It is plausible in the current system design that the quality of the LLM-based planner could impact overall performance. Moreover, **Chameleon** generates the program at one step, without incorporating a re-planning mechanism as the modules in the program are processed. Furthermore, we make the assumption that the list of modules and their descriptions will fit within the context window of LLMs, which may not always be the case. As the task complexity increases and the module inventory expands, there might be a corresponding surge in computational demands or limitations due to the context limit, indicating potential areas for future optimization. However, these potential areas for enhancement don't detract from the paper's central achievements, but instead provide valuable directions for future work and research.

## C  Broader Impacts

The work presented in this paper, **Chameleon**, has significant potential for positive societal impact. By augmenting large language models (LLMs) with plug-and-play modules for compositional reasoning, **Chameleon** can provide more accurate responses to complex, multi-modal tasks, making it a potentially valuable framework for various applications, including but not limited to education, finance, and decision support systems. Additionally, the system's ability to synthesize programs without requiring any training could democratize access to AI technology, enabling non-experts to leverage the power of AI in diverse fields. As research continues to advance in large language models and tool integration, we anticipate that our framework will serve as a foundation for further innovations in pursuing more generalizable and efficient solutions to complex reasoning tasks.

While there might be negative societal impacts associated with the **Chameleon**, such as misinformation and privacy concerns if data sources and external tools it utilizes are not curated meticulously, we believe these risks can be carefully managed and minimized. There's also a risk that excessive reliance on **Chameleon**'s increased autonomy may undermine critical thinking skills or job functions. To effectively mitigate these issues, careful curation of data sources and external tools, along with a strong commitment to user data protection, are essential. Additionally, **Chameleon**'s autonomy should be viewed as a means to augment, not replace, human capabilities. Therefore, the development of robust ethical guidelines, transparency mechanisms, and safeguards is critical, underlying our commitment to the socially responsible deployment of AI.

---

[4]https://www.usgs.gov/geology-and-ecology-of-national-parks/ecology-death-valley-national-park-0

Table 8: The prompt constructed for the planner model on the ScienceQA task. The prompt consists of the instruction that describes the role of the planner model, the in-context examples that map the problem to the module sequence, and the test example.

You need to act as a policy model, that given a question and a modular set, determines the sequence of modules that can be executed sequentially can solve the question.

The modules are defined as follows:

**Program_Generator**: This module generates a Python program that can solve the given question. It takes in the question and possible context and produces a program that can be executed by the "Program_Executor" module. Normally, we consider using "Program_Generator" when the questions and contexts involve complex computation, such as arithmetic operations over multiple numbers, or when the questions involve complex logical operations, such as "if-else" statements.
**Program_Verifier**: This module verifies whether the generated program from "Program_Generator" is valid and error-free. It checks for syntax errors, logical errors, and other potential issues that may arise during program execution.
**Program_Executor**: This module executes the generated program from "Program_Generator" and produces an output that can be further processed by other modules, such as "Question_Answering".
**Row_Lookup**: This module returns the simplified table that only remains the rows that are relevant to the question. It takes in the question and a table and returns the simplified table. If all rows are relevant or there are only three rows or fewer, return the original table. Normally, we only consider using "Row_Lookup" when the table involves more than three rows and the question only requires a small number of rows to answer the question.
**Column_Lookup**: This module returns the simplified table that only remains the columns that are relevant to the question. It takes in the question and a table and returns the simplified table. If all columns are relevant or there are only two columns, return the original table. Normally, we consider using "Column_Lookup" when the table involves more than two columns and the question only requires a small number of columns to answer the question.
**Table_Verbalizer**: This module converts the table to a description that can be easily understood by the downstream modules, like "Program_Generator", "Solution_Generator", "Question_Answering". Normally, we consider using "Table_Verbalizer" when the table involves a small number of rows and columns and the table is domain-specific, such as steam-and-leaf plots, function tables, etc.
**Knowledge_Retrieval**: This module retrieves domain-specific knowledge for the given question and table. Normally, we consider using "Knowledge_Retrieval" when the question and table involve domain-specific knowledge, such as "steam-and-leaf plots", "function tables", "tax forms", etc.
**Solution_Generator**: This module generates a detailed solution to the question based on the information provided. Normally, we use "Solution_Generator" when the question and table involve simple computation, such as arithmetic operations over a single number.
**Answer_Generator**: This module extracts the final answer in a short form from the solution or execution result. This module normally follows the "Solution_Generator" or "Problem_Executor" module.

Below are some examples that map the problem to the modules.

**Table:**
designer watch | $8,141
designer coat | $6,391

**Question:** How much more does a designer watch cost than a designer coat? (unit: $)

**Modules:** `["Program_Generator","Program_Verifier","Program_Executor","Answer_Generator"]`

Table 9: The prompt constructed for the planner model on the TabMWP task. Similarly, the prompt consists of the instruction, the in-context examples, and the test example.

Read the following question, and generate the background knowledge as the context information that could be helpful for answering the question.

**Question:** Which property do these three objects have in common?

**Options:** (A) hard (B) soft (C) yellow

**Metadata:** 'pid': 43, 'has_image': True, 'grade': 4, 'subject': 'natural science', 'topic': 'physics', 'category': 'Materials', 'skill': 'Compare properties of objects'

**Detected text in the image:** ['handkerchief', 'slippers', 'leisure suit']

**Knowledge:**
- This question is about comparing the properties of three objects: a handkerchief, slippers, and a leisure suit.
- The objects are related to the topic of physics and the skill of comparing properties of objects.
- Properties of objects can include physical characteristics such as color, texture, shape, size, weight, and material.

Table 10: The prompt constructed for the "Knowledge Retrieval" module on the ScienceQA task.

Read the following question and metadata, and generate the query for browser search as the context information that could be helpful for answering the question.

**Question:** Which property do these two objects have in common?

**Options:** (A) hard (B) bendable

**Metadata:** 'pid': 329, 'has_image': True, 'grade': 2, 'subject': 'natural science', 'topic': 'physics', 'category': 'Materials', 'skill': 'Compare properties of objects'

**Detected text in the image:** [([[41, 183], [131, 183], [131, 199], [41, 199]], 'rubber gloves'), ([[245, 183], [313, 183], [313, 197], [245, 197]], 'rain boots')]

**Search Query:** Common material properties of jump rope and rubber gloves

Table 11: The prompt constructed for the "Query Generator" module on the ScienceQA task.

> ▷ *Instruction*

Given the question (and the context), select the answer from the options ["A", "B", "C", "D", "E"]. You should give concise and step-by-step solutions. Finally, conclude the answer in the format of "the answer is [ANSWER]", where [ANSWER] is one from the options ["A", "B", "C", "D", "E"]. For example, "the answer is A", "the answer is B", "the answer is C", "the answer is D", or "the answer is E". If the answer is not in the options, select the most possible option.

> ▷ *In-context example(s)*

**Question:** Which property do these two objects have in common?

**Context:** Select the better answer.

**Options:** (A) hard (B) bendable

**Metadata:** 'pid': 6493, 'has_image': True, 'grade': 2, 'subject': 'natural science', 'topic': 'physics', 'category': 'Materials', 'skill': 'Compare properties of objects'

**Image caption:** A pair of scissors next to a pair of scissors.

**Detected text with coordinates in the image:** [([[53, 185], [121, 185], [121, 199], [53, 199]], 'jump rope'), ([[233, 183], [323, 183], [323, 201], [233, 201]], 'rubber gloves')]

**Retrieved knowledge:**
- This question is about comparing the properties of two objects: rubber gloves and rain boots.
- The objects are related to the topic of physics and the skill of comparing properties of objects.
- Properties of objects can include physical characteristics such as color, texture, shape, size, weight, and material. In this case, the two objects have the property of being bendable in common.

**Bing search response:** The most common materials used for disposable gloves are Latex, Vinyl and Nitrile. Each material has its benefits and drawbacks. Latex Gloves are constructed from Natural Rubber Latex and are the most popular type of disposable glove.

**Solution:** An object has different properties. A property of an object can tell you how it looks, feels, tastes, or smells. Different objects can have the same properties. You can use these properties to put objects into groups. Look at each object. For each object, decide if it has that property. A bendable object can be bent without breaking. Both objects are bendable. A hard object keeps its shape when you squeeze it. The rubber gloves are not hard. The property that both objects have in common is bendable. Therefore, the answer is B.

Table 12: The prompt constructed for the "Solution Generator" module on the ScienceQA task.

> ▷ *Instruction*

Read the following table and question, and generate the domain-specific knowledge as the context information that could be helpful for answering the question.

> ▷ *In-context example(s)*

**Table:**
x | y
10 | 15
11 | 9
12 | 2
**Question:** The table shows a function. Is the function linear or nonlinear?

**Knowledge:**
- A linear function is a function whose graph is a straight line.
- A nonlinear function is a function whose graph is not a straight line.
- The equation of a linear function is $y = mx + b$, where $m$ is the slope and $b$ is the $y$-intercept.
- The equation of a nonlinear function is not $y = mx + b$.

Table 13: The prompt constructed for the "Knowledge Retrieval" module on the TabMWP task.

Read the following question and table. Each row is separated by a newline ('\n') and each column is separated by a vertical bar ('|'). Return the simplified table that only remains the rows that are relevant to the question. If all rows are relevant, or the number of rows is fewer than three, return the original table.

**Question:** In preparation for graduation, some teachers and students volunteered for the various graduation committees. How many people are on the music committee?

**Table:**
Committee | Students | Teachers
Program | 5 | 17
Ticket | 20 | 5
Music | 20 | 15
Schedule | 15 | 20
Food | 18 | 2

**Simplified Table:**
Committee | Students | Teachers
Music | 20 | 15

Table 14: The prompt constructed for the "Row Lookup" module on the TabMWP task.

Read the following question and table. Each row is separated by a newline ('\n') and each column is separated by a vertical bar ('|'). Return the simplified table that only remains the columns that are relevant to the question. If all columns are relevant, return the original table.

**Question:** Look at the following schedule. When does Recess end?

**Table:**
Subject | Begin | End
Recess | 6:15 A.M. | 7:20 A.M.
Orchestra | 7:30 A.M. | 8:40 A.M.
Art | 8:45 A.M. | 9:35 A.M.
Handwriting | 9:45 A.M. | 10:20 A.M.
Gym | 10:30 A.M. | 11:15 A.M.
Choir | 11:20 A.M. | 12:25 P.M.
Science | 12:35 P.M. | 1:35 P.M.
Reading | 1:40 P.M. | 2:50 P.M.

**Simplified Table:**
Subject | End
Recess | 7:20 A.M.
Orchestra | 8:40 A.M.
Art | 9:35 A.M.
Handwriting | 10:20 A.M.
Gym | 11:15 A.M.
Choir | 12:25 P.M.
Science | 1:35 P.M.
Reading | 2:50 P.M.

Table 15: The prompt constructed for the "Column Lookup" module on the TabMWP task.

> *▷ Instruction*

Read the following question and table. Write a textual description of the table. The description should keep the critical information in the table for answering the question. The description should not answer the question.

> *▷ In-context example(s)*

**Table:**
Committee | Students | Teachers
Program | 5 | 17
Ticket | 20 | 5
Music | 20 | 15
Schedule | 15 | 20
Food | 18 | 2

**Table description:** The table shows the number of students and teachers on each of the four graduation committees: Program, Ticket, Music, and Schedule. The Music committee has 20 students and 15 teachers.

Table 16: The prompt constructed for the "Table Verbalizer" module on the TabMWP task.

> *▷ Instruction*

Read the following table and then write Python code to answer a question.

> *▷ In-context example(s)*

**Table:**
Price | Quantity demanded | Quantity supplied
$895 | 21,000 | 3,400
$945 | 17,200 | 7,400
$995 | 13,400 | 11,400
$1,045 | 9,600 | 15,400
$1,095 | 5,800 | 19,400

**Questions:** Look at the table. Then answer the question. At a price of $995, is there a shortage or a surplus? Please select from the following options: ['shortage', 'surplus'].

**Code:**
```python
# Python Code, return 'ans'. Make sure that 'ans' is a string selected
    from the options in the question
quantity_demanded_at_price_955 = 13400
quantity_supplied_at_price_955 = 11400
if quantity_demanded_at_price_955 > quantity_supplied_at_price_955:
    ans = 'shortage'
else:
    ans = 'surplus'
```

Table 17: The prompt constructed for the "Program Generator" module on the TabMWP task.

Table 18: The prompt constructed for the "Solution Generator" module on the TabMWP task.

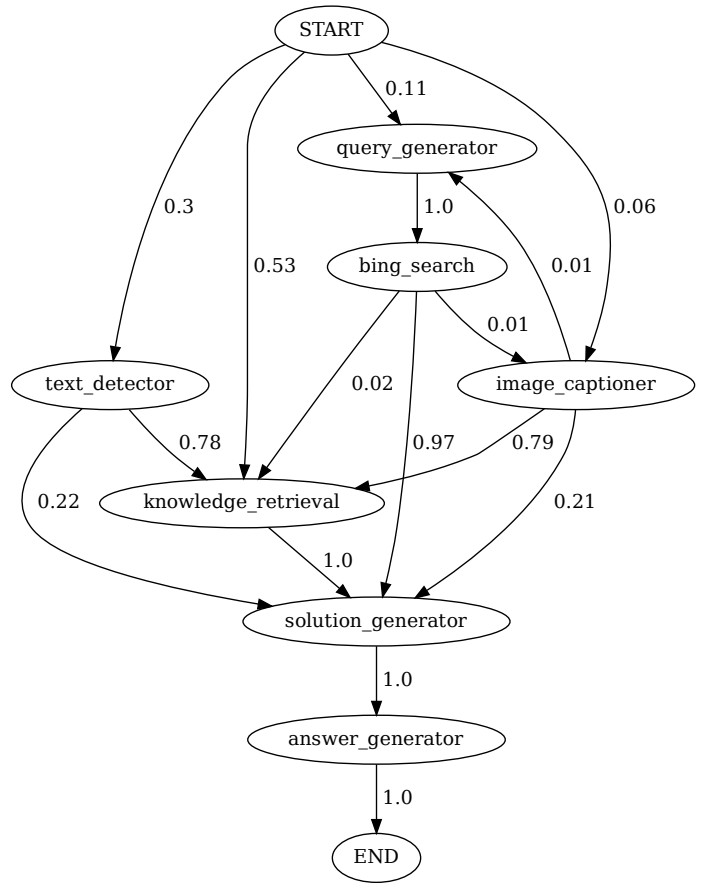

Figure 7: Transitions between modules in programs generated by **Chameleon** (GPT-4) on ScienceQA. START is the start symbol, END is a terminal symbol and the others are non-terminal symbols.

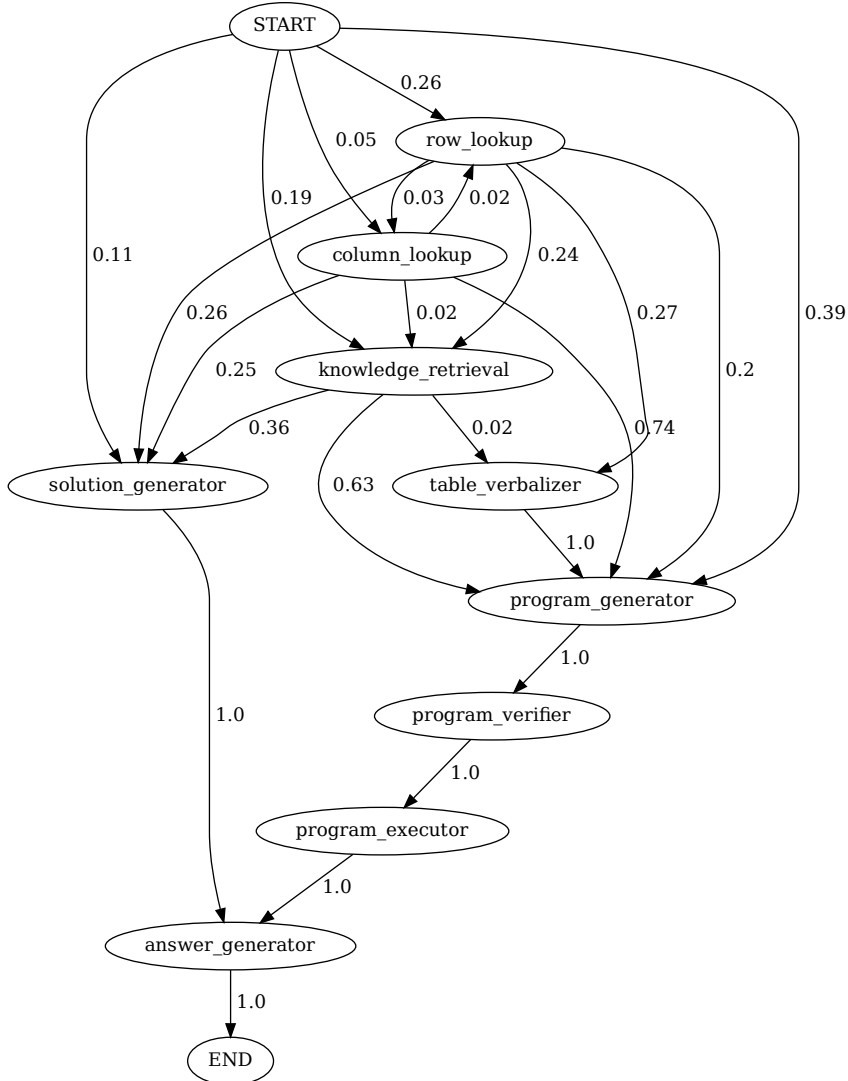

Figure 8: Transitions between modules in programs generated by **Chameleon** (GPT-4) on TabMWP. START is the start symbol, END is a terminal symbol and the others are non-terminal symbols.

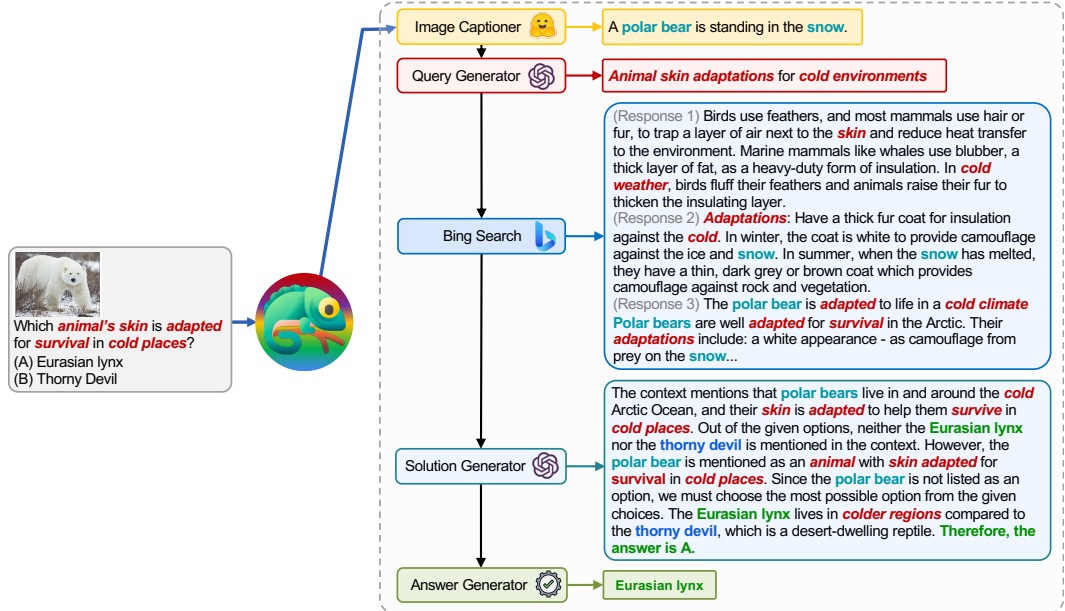

Figure 9: One more example from our **Chameleon** (GPT-4) approach on ScienceQA.

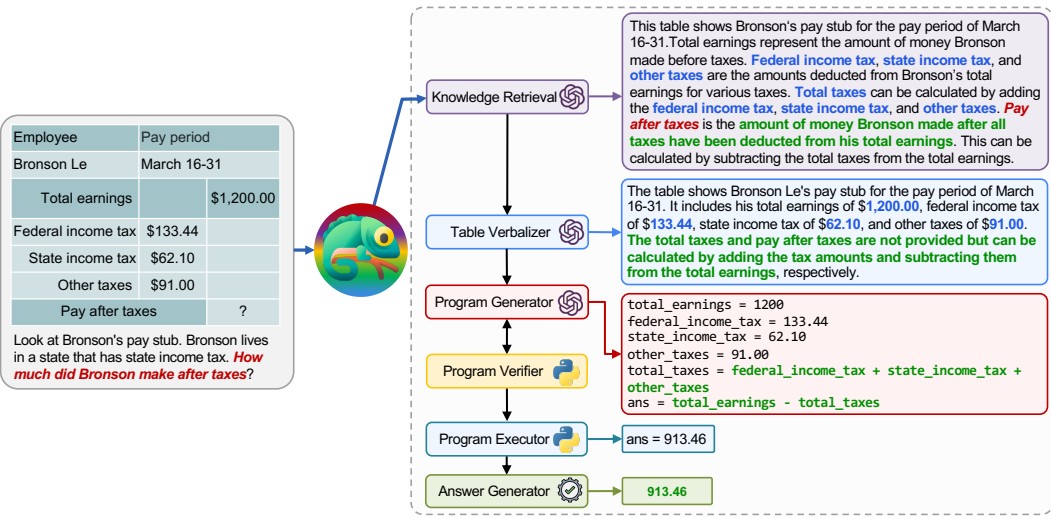

Figure 10: One more example from our **Chameleon** (GPT-4) approach on TabMWP.

> *▷ Query*

**Question:** Which better describes the Death Valley ecosystem?

**Context:** Figure: Death Valley. Death Valley is a desert ecosystem in eastern California.

**Options:** (A) It has dry, thin soil. It also has many different types of organisms. (B) It has a small amount of rain. It also has only a few types of organisms.

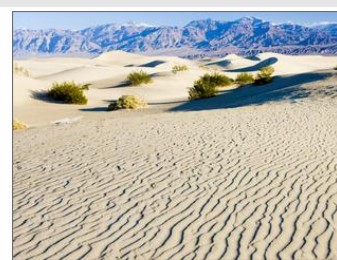

**Metadata:** 'pid': '18973', 'has_image': True, 'grade': 3, 'subject': 'natural science', 'topic': 'biology', 'category': 'Ecosystems', 'skill': 'Describe ecosystems'

> *▷ Response*

**Generated Program:** `["Query_Generator","Bing_Search","Solution_Qenerator","Answer_Generator"]`

**Query Generator:**
Characteristics of Death Valley ecosystem

**Bing Search:** (the first one from response candidates)
Visitors come to Death Valley to experience the stark and lonely vastness of the valley, the panorama of rugged canyons and mountains; the pleasures of the dry, moderate winter climate, the challenge of the hot, arid summer; the relief of the cooler mountains; and the reminders of frontier and Native American ways of life.

**Solution Generator:**
Death Valley is a desert ecosystem, which means it has specific characteristics related to its climate and the organisms that live there. A desert ecosystem typically has dry, thin soil and a small amount of rain. This environment supports only a few types of organisms that are adapted to survive in such harsh conditions. Therefore, the answer is B.

**Answer Generator (Wrong prediction):**
(B) It has a small amount of rain. It also has only a few types of organisms.

**Correct Answer (Ground truth):**
(A) It has dry, thin soil. It also has many different types of organisms.

Table 19: An example of failure cases in ScienceQA from **Chameleon** (GPT-4) where *some modules perform inaccurately*. The generated query from the "Query Generator" module is insufficiently specific, leading to a response from "Bing Search" that does not adequately detail the characteristics of the Death Valley ecosystem, particularly with regard to the types of organisms present. (From U.S. Geological Survey: *Contrary to its name, Death Valley National Park is very much alive ... Despite the low precipitation and extreme summer temperatures, Death Valley is home to many diverse species, several of which are endemic (found nowhere else in the world).*)[4]

<div>

▷ *Query*

**Question:** Which property matches this object?

**Context:** Select the better answer.

**Options:** (A) stretchy (B) yellow

**Metadata:** 'pid': '10338', 'has_image': True, 'grade': 2, 'subject': 'natural science', 'topic': 'physics', 'category': 'Materials', 'skill': 'Identify properties of an object'

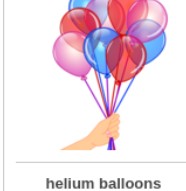

helium balloons

▷ *Response*

**Generated Program:** `["Image_Captioner","Solution_Generator","Answer_Generator"]`

**Image Captioner:**
A colorful kite is being flown in the sky.

**Solution Generator:**
Look at the image of the object, which is a colorful kite. A stretchy object can be stretched without breaking. Kites are typically made of materials that are not stretchy, such as nylon or polyester. A yellow object has a yellow color. The kite in the image is colorful and includes the color yellow. Therefore, the answer is B.

**Answer Generator (Wrong prediction):** (B) yellow

**Correct Answer (Ground truth):** (A) stretchy

</div>

Table 20: An example of failure cases in ScienceQA from **Chameleon** (GPT-4) where *the generated programs are suboptimal*. The LLM planner does not call the "Text Detector" and "Knowledge Retrieval" tools, resulting in an absence of semantic information from the image and a lack of related domain knowledge. Additionally, the employed image captioning model in providing an accurate textual description of the given visual context.

**Question:** Compare the average kinetic energies of the particles in each sample. Which sample has the higher temperature?

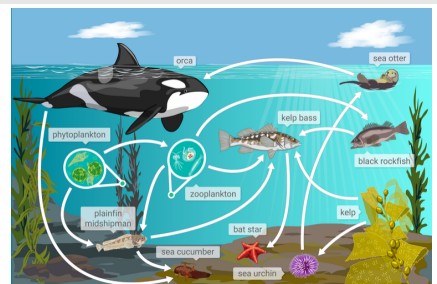

**Context:** Below is a food web from an ocean ecosystem in Monterey Bay, off the coast of California. A food web models how the matter eaten by organisms moves through an ecosystem. The arrows in a food web represent how matter moves between organisms in an ecosystem.

**Options:** (A) bat star (B) phytoplankton (C) kelp bass

**Metadata:** {'pid': '15175', 'has_image': True, 'grade': 6, 'subject': 'natural science', 'topic': 'biology', 'category': 'Ecological interactions', 'skill': 'Interpret food webs II'}

**Generated Program:** ["Text_Detector","Knowledge_Retrieval","Solution_Generator","Answer_Generator"]

**Text Detector:**
[([297, 91], 'orca'), ([653, 85], 'sea otter'),
([487, 191], 'kelp bass'), ([73, 221], 'phytoplankton'),
([625, 269], 'black rockfish'), ([323, 327], 'zooplankton'),
([149, 363], 'plainfin'), ([593, 361], 'kelp'),
([132, 380], 'midshipman'), ([401, 391], 'bat star'),
([271, 433], 'sea cucumber'), ([409, 471], 'sea urchin')]

**Knowledge Retrieval:**
- The question is about a food web in an ocean ecosystem in Monterey Bay, California.
- A food web models how matter eaten by organisms moves through an ecosystem.
- The arrows in a food web represent how matter moves between organisms in an ecosystem.
- The organisms mentioned in the question are bat star, phytoplankton, and kelp bass.
- The target organism is the sea cucumber.
- In a food web, matter moves from one organism to another through consumption, predation, or decomposition.
- Understanding the relationships between organisms in a food web can help determine the flow of matter between them.

**Solution Generator:**
Analyze the food web to determine which organism's matter eventually moves to the sea cucumber.
- The sea cucumber consumes the bat star.
- The bat star consumes the sea urchin.
- The sea urchin consumes the kelp.
- The kelp gets its nutrients from the water, which contains matter from phytoplankton and other sources.
In this food web, the matter from the bat star eventually moves to the sea cucumber through consumption. Therefore, the answer is A.

**Answer Generator (Wrong prediction):** (A) bat star

**Correct Answer (Ground truth):** (B) phytoplankton

Table 21: An example of failure cases in ScienceQA from **Chameleon** (GPT-4) where *the current module inventory falls short in addressing specific abilities*. In this instance, the module inventory does not possess the necessary tools capable of parsing the arrows in the food web illustration.

| ▷ *Query* | |
|---|---|

**Table:**

| x | y |
|---|---|
| 4 | 15 |
| 19 | 18 |
| 11 | 10 |
| 11 | 1 |

**Table:** (semi-structured format)

x | y
4 | 15
19 | 18
11 | 10
11 | 1

**Question:** Look at this table. Is this relation a function?

**Options:** (A) yes (B) no

| ▷ *Response* | |
|---|---|

**Generated Program:** ["Knowledge_Retrieval","Program_Generator","Program_Verifier","Program_Executor","Answer_Generator"]

**Knowledge Retrieval:**
- A relation is a set of ordered pairs $(x, y)$.
- A function is a relation in which each input ($x$-value) is paired with exactly one output ($y$-value).
- To determine if a relation is a function, check if each $x$-value is paired with only one $y$-value.
- If an $x$-value is paired with more than one $y$-value, the relation is not a function.

**Program Generator:**

```
relation = {
    4: 15,
    19: 18,
    11: 10,
    11: 1
}

x_values = list(relation.keys())
unique_x_values = set(x_values)

if len(x_values) == len(unique_x_values):
    ans = 'yes'
else:
    ans = 'no'
```

**Program Verifier: True**

**Program Executor:**

```
ans = 'yes'
```

**Answer Generator (Wrong prediction):** (A) yes

**Correct Answer (Ground truth):** (B) no

Table 22: An example of failure cases in TabMWP from **Chameleon** (GPT-4) where *some modules perform inaccurately*. In this case, the LLM-based program generator is unsuccessful in creating a program that correctly mirrors the mathematical relationship expressed in the provided query and the retrieved domain knowledge.

> *▷ Query*

**Question:** This is Oliver's complete financial record for September. How much money did Oliver have after he purchased the basketball?

**Table:**

| Date | Description | Received | Expenses | Available Funds |
|------|-------------|----------|----------|-----------------|
| | Balance: end of August | | | $154.90 |
| 9/1 | notebook | | $6.95 | $147.95 |
| 9/15 | walking dogs | $15.00 | | $162.95 |
| 9/21 | basketball | | $11.35 | $151.60 |

**Table:** (semi-structured format)
Date | Description | Received | Expenses | Available Funds
| Balance: end of August | | | $154.90
9/1 | notebook | | $6.95 | $147.95
9/15 | walking dogs | $15.00 | | $162.95
9/21 | basketball | | $11.35 | $151.60

> *▷ Response*

**Generated Program:** ["Row_Lookup","Solution_Generator","Answer_Generator"]

**Row Lookup:**
Date | Description | Received | Expenses | Available Funds
9/21 | basketball | | $11.35 | $151.60

**Solution Generator:**
(Step 1) Subtract the cost of the basketball from the available funds before the purchase.
(Step 2) $151.60 - $11.35 = $140.25
(Step 3) Oliver had $140.25 after he purchased the basketball.
(Step 4) The answer is $140.25.

**Answer Generator (Wrong prediction):** 140.25

**Correct Answer (Ground truth):** 151.60

Table 23: An example of failure cases in TabMWP from **Chameleon** (GPT-4) where *the generated programs are suboptimal*. In the given example, the simplified table produced by the "Row Lookup" lacks the structural relations present in the original table's context. These relations are crucial for the LLM-based solution generator to accurately comprehend the domain-specific table.

**Table:**

| Tour bus schedule | | | | |
|---|---|---|---|---|
| historic house | 10:15 A.M. | 10:30 A.M. | 10:45 A.M. | 11:00 A.M. |
| downtown | 11:00 A.M. | 11:15 A.M. | 11:30 A.M. | 11:45 A.M. |
| skyscraper | 11:30 A.M. | 11:45 A.M. | 12:00 P.M. | 12:15 P.M. |
| old building | 12:30 P.M. | 12:45 P.M. | 1:00 P.M. | 1:15 P.M. |
| governor's mansion | 1:00 P.M. | 1:15 P.M. | 1:30 P.M. | 1:45 P.M. |

**Table Title:** Tour bus schedule

**Table:** (semi-structured format)
historic house | 10:15 A.M. | 10:30 A.M. | 10:45 A.M. | 11:00 A.M.
downtown | 11:00 A.M. | 11:15 A.M. | 11:30 A.M. | 11:45 A.M.
skyscraper | 11:30 A.M. | 11:45 A.M. | 12:00 P.M. | 12:15 P.M.
old building | 12:30 P.M. | 12:45 P.M. | 1:00 P.M. | 1:15 P.M.
governor's mansion | 1:00 P.M. | 1:15 P.M. | 1:30 P.M. | 1:45 P.M.

**Question:** This is Oliver's complete financial record for September. How much money did Oliver have after he purchased the basketball?

**Options:** (A) 2 hours and 15 minutes (B) 2 hours and 45 minutes (C) 15 minutes (D) 30 minutes

▷ *Response*

**Generated Program:** `["Row_Lookup","Solution_Generator","Answer_Generator"]`

**Row Lookup:**
historic house | 10:15 A.M. | 10:30 A.M. | 10:45 A.M. | 11:00 A.M.
governor's mansion | 1:00 P.M. | 1:15 P.M. | 1:30 P.M. | 1:45 P.M.

**Solution Generator:**
(Step 1) Find the last departure time from the historic house and the first arrival time at the governor's mansion:
       Historic house last departure: 11:00 A.M.
       Governor's mansion first arrival: 1:00 P.M.
(Step 2) Subtract the departure time from the arrival time:
       1:00 P.M. - 11:00 A.M. = 2 hours
(Step 3) It takes 2 hours to get from the historic house to the governor's mansion.
(Step 4) The answer is 2 hours and 15 minutes.

**Answer Generator (Wrong prediction):** (A) 2 hours and 15 minutes

**Correct Answer (Ground truth):** (B) 2 hours and 45 minutes

Table 24: An example of failure cases in TabMWP from **Chameleon** (GPT-4) where *some modules perform inaccurately and the generated programs are suboptimal*. In this instance, the LLM-based solution generator struggles to understand the bus schedule, which incorporates domain-specific knowledge. Furthermore, the LLM planner does not utilize tools like "Table Verbalizer" and "Column Lookup", which could enhance the LLM's comprehension of the tabular context.

