# OpenReview forum: "Chameleon: Plug-and-Play Compositional Reasoning with Large Language Models"
_NeurIPS.cc/2023/Conference — NeurIPS 2023 poster_

### Official Review · Reviewer_uov2 · 2023-07-02

**Soundness:** 4 excellent
**Presentation:** 2 fair
**Contribution:** 3 good
**Rating:** 4
**Confidence:** 4

**Summary:**

This work presents Chameleon, a system that uses a library of tools or modules to assist a LLM to answer questions. First, a LLM is prompted with a description of available modules, constraints on how the modules should be used, few-shot demonstration examples, and finally the question to answer. The LLM then generates a sequence of modules that should be used to process the question. The modules are used in sequence, keeping track of module outputs and changes to the input question. Finally, the last module outputs an answer to the original question. When Chameleon uses GPT-4 as the base LLM, it achieves +2.55% improvement over GPT-4 with Chain-of-Thought prompting on ScienceQA, and +1.85% improvement over GPT-4 with Program-of-Thought prompting on TabMWP.

**Strengths:**

I see 4 main strengths of this paper:
1. The idea of performing tool use entirely within a sequence of modules (instead of interleaving tool use with prediction from the original LLM) is new and offers some specific advantages
2. Achieving a small performance increase on two question-answering datasets
3. The project being an example of how engineering effort can be organized to achieve high dataset-specific performance
4. The paper writing is well organized and easy to read

Originality: The most original aspect of this work is that the solution is constructed entirely within the modules, whereas previous work on tool use (to my knowledge) interleaves modules with prediction from the original LLM. With the Chameleon approach, the necessary reasoning at each step can be more tightly controlled (e.g., with module-specific prompts and heuristics about what information to include or exclude for future modules), and constraints can be placed on the sequence of modules to use.

Quality: The authors provide an extensive comparison to prior models on two datasets, along with an ablation study for the importance of various modules. The experiments show a slight improvement over previous approaches.

Clarity: The paper is well organized, and the supplemental material contains extensive additional details for the modules.

Significance: The research topic (how to best use LLMs with tools) is definitely significant.

**Weaknesses:**

Overall, this paper feels less scientific and more like a report on an engineering accomplishment in achieving high accuracy on the benchmark datasets, since the modules were created with the datasets in mind, a dataset-specific subset of modules is available at test time, each module includes hand-designed heuristics for how to update the input and cached information, there are hand-designed constraints on module ordering, and dataset-specific “default” module sequences are used if the predicted one does not meet the constraints. All of this points to my interpretation that Chameleon is a framework that can be instantiated, with *sufficient engineering effort*, into a system with high *dataset-specific* accuracy. However, there is little evidence that the Chameleon approach can produce a *single, general-purpose* question answering system with high accuracy on a variety of datasets.

From my perspective, the main weakness is the lack of some general takeaway that can be applied to other scenarios. There are some avenues toward potential takeaways but currently they are not explored thoroughly enough to draw confident conclusions. In the Questions section below, I provide some potential takeaways that could be explored in more detail.

Another weakness of the paper is that the writing oversells the approach in various ways. Please see the Questions section below for specifics. I hope the authors edit the writing accordingly.

Originality: There is not much novelty other than the “sequence of modules” approach mentioned in the Strengths section. Although, if the “sequence of modules” approach is shown to be better than prior approaches through further experiments, then its novelty would be sufficient.

Quality: It is unclear what the main takeaways of the paper should be, or how well-supported they are.

Clarity: The writing oversells the work in a few ways.

Significance: Some of the potential takeaways, if explored more thoroughly, could be quite significant. However, the actual conclusions one can draw from evidence in the paper are less significant right now.

**Questions:**

## Potential takeaways that could be explored in more detail

* Potential takeaway 1: performing tool use entirely within the sequence of modules is better than interleaving tool use with LLM generation as in prior work, e.g., ToolFormer. This might be due to constraints in module ordering, heuristics in how modules update the context, using module-specific prompts instead of the original prompt which might be overloaded, or interleaved tool use “interrupting” the flow of text generation. I would love to see an analysis of these different factors using the same set of tools, for a more general understanding of the benefit of Chameleon’s “sequence of modules” approach.

* Potential takeaway 2: the Chameleon framework is truly general-purpose, meaning the same system achieves high performance on a variety of datasets. This would represent an important step forward in question answering. But currently, we only see dataset-specific variations of Chameleon performing well on one dataset at a time, which is less impressive because the comparison is to GPT-4 with CoT or PoT which has no dataset-specific design or engineering (other than a few-shot prompt). How does Chameleon perform on each dataset with all modules available, a single setting of module ordering constraints, and a single default module sequence? (If the performance is then worse than GPT-4 CoT or PoT, then is Chameleon truly a generally applicable approach?)

* Potential takeaway 3: if the goal is to achieve high accuracy on a single academic dataset, then one can find success using Chameleon with dataset-specific engineering and heuristics. This takeaway is already supported by the paper, but in my opinion, this is a weak result. There are no dataset-specific approaches in the comparison (at least no recent ones building upon ChatGPT or GPT-4). These are also *academic* datasets that are meant to serve as a proxy for measuring more general ability; raw performance on these datasets without generalization to other tasks is not as impressive. On the other hand, if Chameleon could get high performance on a more general and broad dataset, for example (hypothetically) a representative subset of queries to a search engine or AI assistant that are related to science/math/reasoning/etc., then it could be argued that Chameleon is general enough for practical applications, and then this takeaway would carry more weight.

In light of these, I'm not sure what readers are supposed to take away from the paper in its current state, and so I cannot recommend acceptance at this time. However I do think any of the potential takeaways above, if explored thoroughly, could lead to a much more impactful paper, so I encourage the authors to continue their investigations.

## Writing that oversells the approach

* The abstract mentions “improving the best published few-shot result by 11.37%” on ScienceQA and 17.0% on TabMWP, and these numbers are repeated throughout the text. While technically true, these numbers are misleading, simply because the performance of GPT-4 with CoT or PoT hasn’t been published before. Certainly the authors can’t claim any “ownership” of GPT-4 plus CoT or PoT, so they should be treated as prior work even if the authors are the first to measure their accuracy on these datasets. Considering this, it is more honest to say that Chameleon achieves **+2.55% on ScienceQA and +1.85% on TabMWP** over previous approaches without fine-tuning. (MM-COT Large is a fine-tuned model achieving +5.14% over Chameleon on ScienceQA, according to Table 3.) Lines 65-68 are carefully worded to avoid mentioning those bolded numbers and should be more transparent.

* Throughout the text, Chameleon is said to produce “NL-like programs”. Is “*programs*” the most accurate term to use here? These are merely **sequences of modules**, without any other programming feature like loops, conditionals, variable binding, helper functions, and so on. I think the term “program” is overselling the complexity of the module sequence plans.

* What does “plug-and-play” mean? This is a key term used in the paper’s title, but I actually think it hides complexity. To add a new module, one must: (1) implement the core module logic, (2) describe the module in the planner’s prompt, (3) describe constraints about when to use the module with respect to other modules in the planner’s prompt, (4) implement checks for those constraints so invalid plans can be rejected, (5) provide few-shot demonstrations of using the module in the planner’s prompt, (6) decide for which datasets the module should be available, (7) design the module’s update_input and update_cache functions. This seems like quite a bit of work -- an acceptable amount of work, but not quite “plug-and-play”. If easily adding new modules is a key feature of Chameleon, then at the very least, the authors should clearly describe all the steps involved in using a new module, but this is currently missing from the paper.

* Why is MM-COT Large not included in Figure 3(a)? It would be misleading to omit the best approach from the “main baselines”.

* Some modules are always used in a particular order. From Figure 6 and 7 from the appendix, the following modules are always used together: (Query Generator -> Bing Search), (Solution Generator -> Answer Generator), and (Program Generator -> Program Verifier -> Program Executor). Would it make sense to combine each group into a single module? Table 1 lists Chameleon as having >10 tools, but if these tools were combined, the count would be 9. Chameleon is described as using modules in a compositional way, but the number of tools is inflated compared to which compositions are actually used.

## Minor questions and suggestions

* Throughout the text: “**our** Chameleon” is awkward phrasing. It would be more natural to say “our approach Chameleon” or simply “Chameleon”
* Line 97: correcting grammar - “has made tremendous progress and **has stimulated** research in **prompt** learning ...”
* Line 125: extraneous space after Chameleon
* Line 153: please provide examples of update_input and update_cache, since they are hand-designed for each module
* Line 153: Is $M_t$ actually $M_i$? Or $M^t$?
* Line 171 and so on: I didn’t notice any usage of the notation $M_{kr}$ elsewhere, is it necessary?
* Line 233: missing quote before Knowledge Retrieval
* Lines 246 - 248: this sentence is repeated
* Line 251: “in the appendix” is repeated
* Line 296: does the “solution-executor module” mean the “Solution Generator module”?

---

Updated scores after the author-reviewer discussions:
* Soundness: 3 -> 4
* Contribution: 2 -> 3
* Rating ("overall score"): 3 -> 4

**Limitations:**

The limitations (Appendix B) should mention that it takes more wallclock time and compute resources to process questions with modules, especially if those modules themselves contain LLM components. If the use of multiple LLM-based tools really proliferates, then this would be a broader societal impact (Appendix C) through an increase in the amount of energy required to process queries, leading to greater environmental impact.

---

> ### Author Rebuttal · Authors · 2023-08-10
>
> Dear Reviewer,
>
> Thank you so much for your thoughtful review. We are encouraged by your recognition of our paper's originality, extensive experiments, and significant research topic. We appreciate your concerns and your insightful comments to lead to a more impactful paper.
>
> We are committed to addressing the concerns and **engaging further** with you to enhance our paper. Your valuable comments have been addressed as follows:
>
> ### W1: Scientific contributions vs. engineering efforts
>
> Thank you for bringing up your concerns. We'd like to emphasize the scientific contributions as follows:
>
> 1. “Pioneering exploration in the field of LLM agents”: Our work represents one of the first efforts in automatic task planning and execution by the LLM agent. It addresses the inherent limitations of existing LLMs and enhances their capabilities. Many concurrent works in Table 1 are unpublished, underscoring the frontier nature of this research area.
>
> 2. **Uniqueness in modularity**: The modular design allows for a systematic and transferable approach, where individual components can be reused or replaced to adapt to various tasks.
>
> 3. **Blend of science and pragmatism**: While achieving high accuracy on benchmarks required engineering efforts, these efforts were grounded in the scientific exploration of enhancing LLMs for complex tasks. The general capabilities of Chameleon are maintained, and we believe the fine-tuning performed reflects an acceptable practice within ML to show how the system can be optimized. Comprehensive ablation studies and analysis further provide insights for future work.
>
> 4. “Light engineering efforts”: The planner and LLM-based tools are prompted with descriptions and demonstrations. Thanks to the nature of the modular design and textual cache, modules can be designed, updated, or replaced individually. These modules, generated from natural-language programs and executed sequentially, ensure straightforward development.
>
> *For more information, please check out our clarification in the general response.*
>
> ### W2: General vs. dataset-specific framework
>
> 1. **General framework**: Our goal is to provide a general framework that allows domain experts to easily incorporate relevant tools, thereby significantly enhancing task performance. This generalization is focused on a modularized framework that facilitates the easy integration of new tools, rather than a single system applicable to numerous tasks.
>
> 2. **Results on new tasks**: The principles guiding Chameleon's design enable it to function as a single, adaptable system capable of achieving high accuracy across various datasets. We've recorded notable performance gains in tasks not covered in the paper, including an 11.85% gain on FinQA and an 11.39% gain on NER, both benchmarks in financial reasoning.
>
> 3. **Limitations in context length**: At the time of our research, the context length for LLMs like GPT-4 was limited to 8K tokens, necessitating the selection of a dataset-specific subset of modules. Recently, GPT-4 has been scaled to support 16K tokens, and Claude has a 100K token window. These technical advancements enhance Chameleon's ability to plan using a complete set of modules.
>
> ### T1: Analysis of different factors in tool use
>
> Concurrent works, such as UCB-Gorilla and Google-LATM, also favor separating tool planning and execution from LLM generation, and your review insightfully touches on underlying reasons:
>
> 1. Complex reasoning often requires a step-by-step process, and having constraints in module ordering **ensures that the logic unfolds effectively**.
>
> 2. By allowing subsequent modules to benefit from updated inputs and cache, Chameleon can **create context-aware responses**.
>
> 3. By generating the tool planning or calling separately, we can **avoid overloading the prompt** and maintain a more streamlined communication with the tools.
>
> 4.  The interleaved approach might interrupt text generation, while Chameleon **preserves the coherence of the text**.
>
> We appreciate your valuable suggestion and will incorporate an analysis in the revision.
>
> ### T2: Chameleon is general-purpose
>
> **As noted in the global response**, ScienceQA and TabMWP were selected as general-purpose complex reasoning benchmarks because of their unique advantages. As discussed in **W2**, the context limitation of current LLMs necessitates dataset-specific variations. By updating the planner to GPT-4 16K or Claude 100K, Chameleon, Chameleon can accommodate a comprehensive and scale-up module inventory, supporting more general implementation. The variations on different datasets in this paper were tailored to optimize the use of LLM computing resources, not to limit its general applicability.
>
> ### T3: High performance on general datasets
>
> 1. **Dataset-specific approaches in comparisons**: We have compared all approaches available up to the time of submission. For instance, we compare LLaMA-Adapter in Table 3, one of the SOTA models proposed in late April. In our revised paper, we will include comparisons with the latest approaches after submission.
>
> 2. **Generalization to other tasks**: Chameleon is a general framework of compositional reasoning that can achieve high performance on general datasets. Its adaptability is evidenced by an 11.85% gain on FinQA and an 11.39% boost on NER, both benchmarks in financial reasoning.
>
> 3. **ScienceQA and TabMWP are diverse**: *As we claimed in the global response*, ScienceQA and TabMWP were selected due to diverse features and general applicability across questions, domains, contexts, and skills, which require a wide array of tool compositions.
>
> ### Writing suggestions
>
> *Due to space limits, we addressed writing suggestions in the Comment Box below.*
>
> If you believe that our clarifications addressed your concerns about the quality of our work, we wish that you would consider *raising the rating* to reflect that our contributions are properly recognized, which is crucial to us.

---

> ### Author Response · Authors · 2023-08-10
> **Responses to writing improvement suggestions**
>
> Thank you for pointing out the valuable suggestions for our paper. We will incorporate them in our revised paper:
>
> ### Result presentation
>
> Our intention was to highlight the improvements of Chameleon over the benchmarks **available at the time**. We understand that this may have created confusion. We appreciate the insight that these comparisons should be treated as prior work. We'll modify our description to state that Chameleon (GPT-4) achieves +2.55% on ScienceQA and +1.85% on TabMWP over previous approaches without fine-tuning. Additionally, we will make adjustments in Lines 65-68 to provide a more straightforward representation of the data.
>
> ### NL-like programs
>
> Our intention in using this term was to highlight that the generated sequences resemble the structure of a Python list, and its simplicity in natural language, aligning with the tool's name. We appreciate your insight and will consider using a more precise term such as “module sequence plans”, or provide an explanation to avoid confusion.
>
> ### Plug-and-play
>
> We agree that the process of adding a new module involves several steps, as you have outlined and we have discussed in the readme file of the codes. The term was intended to highlight the idea that, a new module can be easily integrated within the existing framework without needing to modify other components, compared to existing works. We appreciate your suggestion and will provide a section outlining the steps in the revision, thereby providing clarity and transparency about the process.
>
> ### MM-COT Large
>
> MM-COT Large boasts a much larger tuned parameter size​​ of 738M, while the dominant fine-tune models only have tuned parameter sizes from 1.2M to 223M. We did include MM-COT Large in Table 3 for a comprehensive comparison, but it was excluded in Figure 3(a) to focus on comparisons within the dominant settings. Your feedback is valuable, and in the revision, we will include MM-COT Large in Figure 3(a) or provide a clear explanation.
>
> ### Modules in order
>
> tend to be used together due to the sequential nature of the tasks they perform. However, maintaining them as distinct modules has specific advantages:
>
> 1. **Flexibility and extensibility**: By maintaining module separation, each can be individually updated, improved, or replaced without affecting the others. For example, Solution Generator could be updated with fine-tuned language models, while Answer Generator can adapt to varied formats in new tasks. This modularity is pivotal for future refinements or extensions of Chameleon.
>
> 2. **Clarity in functionality**: Each module has a unique role in the overall reasoning process. Keeping them separate aids in understanding, debugging, and modifying, especially for those unfamiliar with our system's details.
>
> ### Limitation statement
>
> We acknowledge the extended time and compute resources required when using modules with LLM components. This insight will be detailed in Appendix B. Furthermore, the broader societal implications, especially the environmental impact of increased energy consumption, are crucial. We'll address these concerns and potential mitigations in Appendix C.

---

> > ### Comment · Reviewer_uov2 · 2023-08-16
> >
> > Thank you for the detailed rebuttal.
> >
> > However, I still feel that the main takeaway is unclear and not well supported. I don't think this can be resolved with only a rebuttal; I believe new experiments are needed to fully support whatever takeaway the authors wish to communicate.
> >
> > For example, Potential Takeaway 1 was: "performing tool use entirely within the sequence of modules is better than interleaving tool use with LLM generation as in prior work, e.g., ToolFormer". I wrote down a few potential reasons this might be, and suggested the authors perform "an analysis of these different factors", e.g., detailed ablation studies to measure the impact of each potential reason. The rebuttal elaborates on the reasons with words, without any deeper analysis.
> >
> > Potential Takeaway 2 ("the Chameleon framework is truly general-purpose, meaning the same system achieves high performance on a variety of datasets") might take less work to support with experimental results. The authors claim that, with GPT-4 recently supporting a context window of 16K tokens, Chameleon can support all of its modules in a dataset-agnostic way, and the authors also mention new positive results on FinQA and NER. However, no experimental results are provided to verify whether Chameleon **maintains improved performance on all datasets when set up in a dataset-agnostic way**. This is the crucial missing experiment that would support Potential Takeaway 2.
> >
> > I would like to clarify that I don't think it's necessary for one paper to explore _all_ of the potential takeaways. It is more important that _at least one takeaway_ is clearly communicated and very well supported with experimental evidence. But given the current results, I don't think any of those potential takeaways are sufficiently supported yet, so I encourage the authors to continue the work with a focus on demonstrating (at least) one message very well.
> >
> > My scores are unchanged, because the main weakness (lack of a clear well-supported takeaway message) remains.

---

> > > ### Author Response · Authors · 2023-08-18
> > > **Takeaways we claimed in the paper**
> > >
> > > Dear Reviewer,
> > >
> > > Thank you so much for your insightful comments and continued feedback. We would like to clarify **the takeaways we have claimed in our submitted paper, backed by extensive experimental results**.
> > >
> > > **Our takeaway 1: Augmented LLMs demonstrate flexibility and superior performance in complex reasoning tasks.**
> > >
> > > Our approach offers a distinct novelty. It not only decomposes tool planning and execution but also features a modular implementation of the framework. This offers advantages in scalability, adaptability, and ease of integration. Different from concurrent works that focus on runnable demos, we are among the pioneering efforts that examine their performance benefits in compositional reasoning contexts, as recognized by Reviewer yhWh. The performance benefits are eviendenced by comparison with latest few-shot and fine-tuned models, listed in Table 3 and 4.
> > >
> > > **Our takeaway 2: GPT-4 exhibits a more precise and consistent tool planning capability than ChatGPT.** We have investigated GPT-4's planning capabilities, emphasizing its superiority over ChatGPT. These findings are supported by experimental results showcased in Figures 4-7 and Table 7.
> > >
> > > **Our takeaway 3:  Our additional research illuminates potential directions for future developments in LLM agents and augmented LLMs.** The ablation study in Table 5 shows the significance of different modules for complex reasoning tasks. Transition graphs in Figures 6 and 7 study the planning behavior of LLMs on program planning. Through examples presented in Figures 1-2, 8-9, Tables 19-24, and Figure 10 (added in [pdf](https://openreview.net/attachment?id=bIr8g5rvPm&name=pdf) during the rebuttal period), we demonstrate how Chameleon facilitates complex reasoning and identify scenarios where it might fail, providing a roadmap for future research.
> > >
> > > Warm regards,
> > >
> > > The authors of Paper 2615

---

> > > ### Author Response · Authors · 2023-08-18
> > > **Responses to your remaining concerns on potential takeaways**
> > >
> > > Dear Reviewer,
> > >
> > > Thank you so much for your great efforts in reviewing our paper. We highly appreciate your new feedback and would like to address them as follows:
> > >
> > > We would like to clarify that **we didn’t claim these three potential takeaways as you pointed out in our paper submission**. However, we genuinely appreciate the insightfulness of these potential takeaways and would like to delve deeper into them with **both existing and new experiments**.
> > >
> > > ### Potential takeaway 1
> > >
> > > **In our submitted paper**, we **did not claim** that “performing tool use entirely within the sequence of modules is better than interleaving tool use with LLM generation as in prior work, e.g., ToolFormer”. As noted in the ToolFormer paper, ToolFormer generates over 300K data to fine-tune the LLM and only calls one API for every query. This approach is not suitable for complex reasoning problems that necessitate the combination of several different tools, especially when no extensive training data is available. Moreover, it lacks efficiency; for instance, when calling N APIs, the output from the first API would be redundantly fed to the LLM N-1 times. This paradigm, which interleaves tool use with LLM generation, has not been favored in subsequent works as shown in Table 1. Given theses limitations and inefficiencies, a direct experimental comparison between ToolFormer and Chameleon is not currently practical. **We genuinely hope for your understanding on this matter and seek further guidance on whether you'd still like us to conduct such experiments.**
> > >
> > > ### Potential takeaway 2: the Chameleon framework is truly general-purpose
> > >
> > > We're grateful for your suggestion and have implemented Chameleon in a **dataset-general** manner, as you recommended. Specifically, when assessing various datasets, both the planner and tool inventory remain consistent, including task description, tool set, and demonstrations. As evidenced by the results presented below, **Chameleon, when deployed in a dataset-general fashion, maintains similar performance improvements over current SOTA baselines**.
> > >
> > > New results on 1,000 test examples of TabMWP:
> > >
> > > |                  | **Implementation** | **Accuracy (%)** | **Gain over SOTA** |
> > > | ---------------- | ---------------------- | -------------------- | ---------------------- |
> > > | GPT-4 CoT        | dataset-customized     | 90.9                 | /                      |
> > > | GPT-4 PoT        | dataset-customized     | 96.9                 | /                      |
> > > | Chameleon (ours) | dataset-customized     | **98.8**         | **+1.9%**          |
> > > | Chameleon (ours) | **dataset-general**    | **98.5** (new)   | **+1.7%** (new)    |
> > >
> > > New results on 1,000 test examples of ScienceQA:
> > >
> > > |                  | **Implementation** | **Accuracy (%)** | **Gain over SOTA** |
> > > | ---------------- | ---------------------- | -------------------- | ---------------------- |
> > > | GPT-4 CoT        | dataset-customized     | 82.6                 | /                      |
> > > | Chameleon (ours) | dataset-customized     | **84.9**         | **+2.3%**          |
> > > | Chameleon (ours) | **dataset-general**    | **84.8** (new)   | **+2.2%** (new)    |
> > >
> > > Additionally, it is a common optimization strategy to employ task-specific and query-specific adaptations to enhance both performance and API-calling efficiency when developing few-shot LLMs. This approach is supported by multiple papers as follows:
> > > - Learning to retrieve prompts for in-context learning, NAACL22
> > > - Automatic Chain of Thought Prompting in Large Language Models, ICLR23
> > > - Complexity-Based Prompting for Multi-Step Reasoning, ICLR23
> > > - Eeast-to-Most Prompting Enables Complex Reasoning in Large Language Models, ICLR23
> > >
> > > ### Potential takeaway 3: Chameleon can achieve high performance on general datasets
> > >
> > > As the results below indicate, Chameleon achieves an improvement ranging from 1.85% to 11.85% on **four challenging reasoning datasets** compared to strong GPT-4 based baselines. It is therefore evident that Chameleon has the potential to be generalized to broader datasets.
> > >
> > > |           | **ScienceQA**  | **TabMWP**     | **FinQA (New)** | **NER (New)**   |
> > > | --------- | ------------------ | ------------------ | ------------------- | ------------------- |
> > > | GPT-4 CoT | 83.99              | 90.81              | /                   | /                   |
> > > | GPT-4 PoT | /                  | 96.93              | 62.87               | 56.71               |
> > > | Chameleon | **86.54 (+2.55%)** | **98.78 (+1.85%)** | **74.72 (+11.85%)** | **68.10 (+11.39%)** |
> > >
> > > We hope that these clarifications, combined with literature references and additional experimental evidence, validate our takeaways and address your concerns about potential takeaways. **We will incorporate all these clarifications and new experiments into our revised paper.** We are deeply grateful for your invaluable comments and continued feedback, which have been instrumental in enhancing our paper!
> > >
> > > Warm regards,
> > >
> > > The authors of Paper 2615

---

> > > ### Author Response · Authors · 2023-08-21
> > > **New results with Claude and the main takeaway**
> > >
> > > Dear Reviewer,
> > >
> > > We have some new results regarding our Chameleon approach for the updates.
> > >
> > > We switched the base LLMs to **Claude** and observed similar performance improvements. Our proposed method achieves an accuracy gain of 2.1% on ScienceQA and a gain of 15.7% on TabMWP, compared to SOTA few-shot approaches such as CoT and PoT.
> > >
> > > The new results using **Claude** as the base LLM:
> > >
> > > |                              | ScienceQA        | TabMWP            |
> > > | - | - | - |
> > > | CoT (new)             | 78.7             | 74.0              |
> > > | PoT (new)             | /                | 79.6              |
> > > | **Chameleon** (new) | 80.8 **(+2.1%)** | 94.9 **(+15.7%)** |
> > >
> > > The results using **ChatGPT** as base LLM:
> > >
> > > |                         | ScienceQA (%)    | TabMWP (%)       | NER (F1)         |
> > > | - | - | - | - |
> > > | CoT             | 76.9             | 83.3             | /                |
> > > | PoT             | /                | 89.3             | 51.3             |
> > > | **Chameleon** | 78.9 **(+2.0%)** | 94.2 **(+5.1%)** | 68.1 **(+16.8)** |
> > >
> > > The results using **GPT-4** as base LLM:
> > >
> > > |                       | ScienceQA (%)    | TabMWP (%)       | FinQA (%)         | NER (F1)         |
> > > | - | - | - | - | - |
> > > | CoT             | 82.6             | 90.9             | /                 | /                |
> > > | PoT             | /                | 96.9             | 62.9              | 56.7             |
> > > | **Chameleon** | 84.9 **(+2.3%)** | 98.8 **(+1.9%)** | 74.7 **(+11.8%)** | 66.7 **(+11.0)** |
> > >
> > > The **main takeaway from our paper** is that **augmented LLMs using the Chameleon framework can demonstrate flexibility and superior performance in complex compositional reasoning tasks**. This is validated through extensive experiments and analysis, as follows:
> > > 1. Experiments on TabMWP and SciencenQA using ChatGPT and GPT-4 as base LLMs, and comparisons with SOTA approaches (Table 1 and Table 2). Our proposed method achieves a 1.9%-5.1% improvement.
> > > 2. New experiments on two financial reasoning benchmarks, FinQA and NER, show similar substantial improvements with an accuracy gain of 1.9%-5.1% and an F1 score gain of 11.0-16.8, achieved by our method.
> > > 3. New experiments using Claude as the base LLM. Our approach demonstrates consistent improvements across two tasks.
> > > 4. New error analysis in [pdf](https://openreview.net/attachment?id=bIr8g5rvPm&name=pdf) indicates that our proposed method significantly reduces the errors made by base LLMs due to their inherent limitations in using external tools.
> > > 5. The ablation study presented in Table 5 underscores the importance of tools with specific functions for complex reasoning tasks.
> > > 6. The study of tool-use planning (Figure 4 and 5) reveals that GPT-4 possesses superior tool-planning reasoning compared to ChatGPT.
> > > 7. Quantitative studies in Table 7, and Figures 6 and 7 show that the LLM planner can effectively decide how to sequence tools in a few-shot setup.
> > >
> > > We sincerely hope that our new experiments and clarifications address your remaining concerns. Thank you very much for your consideration!
> > >
> > > Best regards,
> > >
> > > The authors of Paper 2615

---

> > > > ### Comment · Reviewer_uov2 · 2023-08-21
> > > >
> > > > The new results are very nice. I am glad to see that the improvements in the initial paper were not due to dataset-specific choices, since the improvements remain mostly the same after making the approach dataset-general. This is strengthened by positive results on new benchmarks and demonstrating improvements using different base LLMs.
> > > >
> > > > ---
> > > >
> > > > I want to be really clear: through my emphasis on "takeaways", I am _not_ directly asking the authors to perform any particular experiment; instead, I am identifying _potential_ takeaways and how those might be supported, in an effort to help the authors strengthen the paper's message. It is up to the authors to choose what key message(s) their paper should communicate. I only want to make sure that those takeaways are clearly communicated and well-supported with evidence.
> > > >
> > > > I'd like to briefly comment on the takeaways that the authors do claim:
> > > >
> > > > > Our takeaway 1: Augmented LLMs demonstrate flexibility and superior performance in complex reasoning tasks.
> > > >
> > > > In my opinion, this is just saying "tool use is powerful" which is well-known at this point. This is not really a new discovery and thus doesn't carry much significance.
> > > >
> > > > > Our approach offers a distinct novelty. It not only decomposes tool planning and execution but also features a modular implementation of the framework.
> > > >
> > > > Do the authors want to highlight, as a _key contribution_, their software engineering design which makes the system easily extendable? If so, more details about the engineering would be needed. What clever design decisions were made and why? Practical evidence such as "a colleague who did not contribute to the original code base took X amount of time and Y lines of code changes to add a new module" would also help.
> > > >
> > > > > Our takeaway 2: GPT-4 exhibits a more precise and consistent tool planning capability than ChatGPT.
> > > >
> > > > I'm really not surprised because GPT-4 is much better than ChatGPT in many ways. Like "Our takeaway 1", I don't think this one carries much significance either.
> > > >
> > > > > Our takeaway 3: Our additional research illuminates potential directions for future developments in LLM agents and augmented LLMs.
> > > >
> > > > It was not clear to me how this paper illuminates clever new research directions that were not known before.
> > > >
> > > > > The main takeaway from our paper is that augmented LLMs using the Chameleon framework can demonstrate flexibility and superior performance in complex compositional reasoning tasks.
> > > >
> > > > In my words, "Chameleon is dataset-general and performs well with several base LLMs". Considering the new results, I think this is now the strongest takeaway and is well supported. I am quite happy about this.
> > > >
> > > > ---
> > > >
> > > > Due to the new results, I am updating my scores. My reasoning is:
> > > >
> > > > * From my initial review, **the main weakness has been remedied**. Specifically, "Chameleon is dataset-general and performs well with several base LLMs" is a clear takeaway message and is supported with the new results.
> > > > * My **new main concern** is simply about the writing being updated. The authors have worked hard to perform new experiments with (1) a new dataset-general setup, (2) new base LLMs, and (3) new datasets. These new experiments are _crucial_ to supporting the paper's message (i.e., they aren't just going into an appendix). Certainly, the paper's main text requires a significant amount of updating:
> > > >   * Updating the description of the method which used to be dataset-specific but is now dataset-general
> > > >   * Providing background on the new base LLMs
> > > >   * Providing background on the new datasets, and discussing/analyzing prior works that targeted those datasets
> > > >   * General reframing or clarification of the paper's message
> > > >   * Any other revisions/clarifications according to all of the author-reviewer discussions
> > > > * Who will review all of the updated text which in total would seem to be a "major revision"? It is unfortunate that the authors are not allowed to upload a revised paper during the discussion period. **I am hesitant to recommend acceptance knowing that the paper will have major changes that are completely unreviewed** (and I think back my concerns about the authors' initial writing, as mentioned in my initial review).
> > > >
> > > > Initial scores:
> > > > * Soundness: 3 good
> > > > * Presentation: 2 fair
> > > > * Contribution: 2 fair
> > > > * Rating: 3 reject
> > > >
> > > > Updated:
> > > > * Soundness: 4 excellent
> > > > * Presentation: 2 fair (I expect this would go up if I were to review a revised paper)
> > > > * Contribution: 3 good
> > > > * Rating: 4 borderline reject (I expect this would go up if I were to review a revised paper)
> > > >
> > > > To the authors: regardless of the accept/reject decision, the work is now _much improved_, and I hope the authors feel that the review process has been helpful.

---

> > > > > ### Author Response · Authors · 2023-08-21
> > > > > **Appreciation for Your Helpful Feedback Throughout the Rebuttal**
> > > > >
> > > > > Dear Reviewer uov2,
> > > > >
> > > > > We are sincerely grateful for your invaluable comments and suggestions throughout the reviewing and rebuttal process.
> > > > >
> > > > > It's encouraging to know that our new experiments have addressed your concerns regarding the generality of Chameleon. **We will incorporate all the new findings into our revised paper and refine it based on the writing suggestions from you and other reviewers**. We believe these updates will make our paper more impactful and appeal to a broader audience in the field. Although we can't update the paper at this moment, **we will release it publicly, and both you and the community are welcome to provide further feedback on our work**.
> > > > >
> > > > > The rebuttal process has been very helpful and fruitful for us! Thank you so much for your significant contribution to our paper.
> > > > >
> > > > > Best regards,
> > > > >
> > > > > The authors of Paper 2615

---

> ### Author Response · Authors · 2023-08-15
> **Thanks for Your Valuable Reviews: Kind Request for Reevaluation**
>
> Dear Reviewer,
>
> Firstly, we wish to express our gratitude for your thorough review of our paper. We genuinely appreciate the time and expertise you dedicated to its evaluation. We have taken each of your comments to heart and addressed them in our rebuttal.
>
> It would be immensely beneficial if you could reconsider our revised arguments and clarifications. We are hopeful that our revisions have addressed your concerns satisfactorily and might warrant **a reevaluation of its acceptance**.
>
> Your feedback has been, and continues to be, pivotal to our work. We would highly value any additional insights or clarifications you might offer.
>
> Thank you once more for your time and consideration.
>
> Sincerely,
>
> The authors of Paper 2615

---

### Official Review · Reviewer_GU39 · 2023-07-05

**Soundness:** 4 excellent
**Presentation:** 4 excellent
**Contribution:** 3 good
**Rating:** 7
**Confidence:** 4

**Summary:**

The paper presents Chameleon, an AI system that enhances large language models (LLMs) with plug-and-play modules for compositional reasoning. Chameleon uses various tools, including LLMs, vision models, web search engines, and Python functions, to perform complex reasoning tasks. The system, powered by GPT-4, significantly improves performance on two multi-modal reasoning tasks: ScienceQA and TabMWP, setting new state-of-the-art results. The authors also demonstrate that Chameleon's GPT-4-powered planner exhibits more consistent and rational tool selection compared to a ChatGPT-powered planner.

**Strengths:**

1. The paper is overall well-written and easy to follow. Abundant figures and visualizations also help with reader understanding.
2. The paper presents a novel system, Chameleon, that enhances large language models (LLMs) with plug-and-play modules for compositional reasoning. This innovative approach addresses the inherent limitations of LLMs, such as their inability to access up-to-date information, use external tools, or perform precise mathematical and logical reasoning.
3. Chameleon achieves impressive performances (especially with GPT-4) on two QA datasets: ScienceQA and TabMWP, showing the effectiveness of the entire framework.
4. Very detailed and thorough analysis of existing related works, which can largely help the community to explore the direction.

**Weaknesses:**

1. I might miss something. However, the novelty of the method seems to be limited. As a method focusing on compositional tool use, I am wondering what are the differences between Chameleon and HuggingGPT (https://arxiv.org/pdf/2303.17580.pdf), except for the application tasks and involved tools. As both works seem to leverage human-introduced ordering information between tools to compositionally use the tools.
2. Both datasets seem to have strong dependencies between different tools. With a relatively small number of simple tools (compared with vision models and SQL interpreters, etc.), the compositional order among tools seems to be quite static and rigid.

**Questions:**

Overall a good paper. No additional questions.

**Limitations:**

The authors have thoroughly demonstrated the limitations of the work in the appendix.

---

> ### Author Rebuttal · Authors · 2023-08-09
>
> Dear Reviewer,
>
> Thank you for your comprehensive and positive remarks about our work. Your recognition of Chameleon's novel approach, addressing the inherent limitations of LLMs reaffirms the value of our work. Your commendation of the achievements emphasizes the effectiveness of our framework. Your appreciation for our detailed examination of existing works, and their potential to guide the community in exploring this direction, is sincerely valued.
>
> We are inspired by your feedback and look forward to **engaging further** with you to refine our paper. Your thoughtful comments have been addressed as follows:
>
> ### W1: Novelty of Chameleon; differences between Chameleon and HuggingGPT
>
> We would like to highlight the **novelty** of Chameleon: (1) It is one of the first works in the field of LLM agents for task planning and execution, as recognized by Reviewer yhWh (R1); (2) It has the potential to be a general framework for compositional reasoning and is distinct from prior works in constructing a solution entirely within modules with natural-language-format program generation, as highlighted by Reviewer uov2 (R4). *For more information, please check out our clarification in the general response.*
>
> We also appreciate the chance to clarify the **differences** between Chameleon and HuggingGPT. It's vital to note that *HuggingGPT is concurrent with our work and is not published; its latest version is published on 25, May, later than the NeurIPS deadline*, further emphasizing the independent innovation in Chameleon:
>
> 1. **Tool composition mechanism**: Chameleon introduces a unique planner-driven approach that dynamically selects and composes tools based on the context of the given task. The planning algorithm accounts for not only the ordering of tools but also the specific context, input, and required reasoning. In contrast, HuggingGPT parses the user request into a task list and assigns appropriate models for each task.
>
> 2. **Task flexibility**: Chameleon is designed to handle a broader array of tasks by employing various tools, such as LLMs, vision models, web search engines, and Python functions. This combination allows Chameleon to perform complex reasoning tasks, such as mathematical reasoning and science QA, beyond the visual understanding tasks that have been demonstrated in HuggingGPT.
>
> 3. **Tool type**:  Chameleon is designed as a general framework, which supports Github models, Huggingface models, Python tools, web engines, and heuristic tools. It has a novel mechanism to update the inputs and cache during the sequential execution of different types of tools. In contrast, HuggingGPT only supports models in Hugging Face, with standard model configurations.
>
> 4. **Plug-and-play nature**: Chameleon features a modular nature where each tool can be replaced and updated individually. The tool can be LLM-based, pre-trained neural networks, or software tools. A new tool is easily included in the module inventory by providing the tool description and demonstration examples to the planner prompt. This makes Chameleon more robust and flexible in handling unseen or complex scenarios.
>
> 5. **Empirical validation**: Chameleon has been empirically validated on two challenging reasoning tasks, setting new state-of-the-art results. Instead, HuggingGPT implements runnable examples and focuses on case studies and qualitative results.
>
> We believe these key differences underscore the novelty of Chameleon. We will enhance the related work section to clarify the distinctions between Chameleon and other methods.
>
> ### W2: The compositional order among tools seems to be static
>
> We appreciate your observation and would like to provide clarifications.
>
> 1. **Tool interdependencies**: While it's true that there are dependencies between different tools, this design **reflects the complex nature of real-world reasoning tasks**. In practice, many reasoning processes require a series of interdependent steps, and our system is intended to mirror these real-world scenarios. As shown in Table 7 in the Appendix, there are 14 and 11 different compositions for Chameleon (ChatGPT) and Chameleon (GPT-4) in ScienceQA, and 28 and 19 different compositions respectively on TabMWP. The diverse dependencies between tools add richness to the reasoning process, allowing Chameleon to leverage various skills.
>
> 2. **Compositional order**: We analyzed the compositional order among tools by visualizing the transition graphs between modules in programs generated by Chameleon (please see Figure 6 and Figure 7 in the Appendix). According to the transition graphs, **the compositional order is dynamic and extensible, rather than static and rigid**. The complex graphs reflect the diverse reasoning paths required for the tasks we address. Interestingly, some frequent paths (compositional orders) in graphs indicate that our planners can find consistent and rational patterns for solving specific types of problems. These paths provide insight into the reasoning strategies employed by the planners, further showcasing the system's ability to execute structured reasoning.
>
> 3. **Scalability and flexibility**: Despite the relatively small number of tools in the presented work, Chameleon is designed to be scalable and can be extended with more complex tools, such as other vision models and SQL interpreters. The current selection of tools serves as a demonstration of the framework's capabilities, and we plan to explore more compositions in future works.
>
> Based on your feedback, we propose to include a detailed explanation in the paper. Thank you once again for your thoughtful comments.

---

> > ### Comment · Reviewer_GU39 · 2023-08-13
> > **Thanks for the response**
> >
> > Thanks the authors for their detailed rebuttal. I have no further questions and I will maintain my positive score.

---

> > > ### Author Response · Authors · 2023-08-13
> > > **Thanks for your positive feedback**
> > >
> > > Dear Reviewer,
> > >
> > > Thank you very much for your thoughtful review and your positive assessment of our work. We appreciate the time and effort you have invested in this process, and we're glad to hear that our rebuttal has addressed your questions satisfactorily.
> > >
> > > Should you have any further questions in the future, or if there are any additional aspects that need clarification, please do not hesitate to reach out to us.
> > >
> > > Sincerely,
> > >
> > > The authors of Paper 2615

---

> > > ### Author Response · Authors · 2023-08-18
> > > **Kind request for review of our new experiments and clarifications**
> > >
> > > Dear Reviewer,
> > >
> > > We greatly appreciate the time and effort you dedicated to reviewing our paper and offering insightful comments. During the rebuttal period, we added **multiple new experiments**. We kindly draw your attention to these updates and eagerly anticipate any further feedback. Your insights are invaluable to refining our paper!
> > >
> > > ### Experiment 1: The performance of Chameleon in a dataset-general way
> > >
> > > We added experiments on Chameleon implemented in a dataset-general manner, as suggested by Reviewer uov2. In this setup, when assessing different datasets, both the planner and tool inventory remain consistent, including task description, tool set, and demonstrations. The results below indicate that **Chameleon, when deployed in a dataset-general fashion, maintains similar performance gains over SOTA baselines**.
> > >
> > > New results on TabMWP:
> > >
> > > |                  | **Implementation** | **Accuracy (%)** | **Gain over SOTA** |
> > > | ---------------- | ---------------------- | -------------------- | ---------------------- |
> > > | GPT-4 CoT        | dataset-customized     | 90.9                 | /                      |
> > > | GPT-4 PoT        | dataset-customized     | 96.9                 | /                      |
> > > | Chameleon (ours) | dataset-customized     | 98.8         | +1.9%          |
> > > | Chameleon (ours) | **dataset-general**    | **98.5** (new)   | **+1.7%** (new)    |
> > >
> > > New results on ScienceQA:
> > >
> > > |                  | **Implementation** | **Accuracy (%)** | **Gain over SOTA** |
> > > | ---------------- | ---------------------- | -------------------- | ---------------------- |
> > > | GPT-4 CoT        | dataset-customized     | 82.6                 | /                      |
> > > | Chameleon (ours) | dataset-customized     | 84.9         | +2.3%          |
> > > | Chameleon (ours) | **dataset-general**    | **84.8** (new)   | **+2.2%** (new)    |
> > >
> > > ### Experiment 2: The performance of Chameleon on general datasets
> > >
> > > |           | **ScienceQA**  | **TabMWP**     | **FinQA (New)** | **NER (New)**   |
> > > | --------- | ------------------ | ------------------ | ------------------- | ------------------- |
> > > | GPT-4 CoT | 83.99              | 90.81              | /                   | /                   |
> > > | GPT-4 PoT | /                  | 96.93              | 62.87               | 56.71               |
> > > | Chameleon | **86.54 (+2.55%)** | **98.78 (+1.85%)** | **74.72 (+11.85%)** | **68.10 (+11.39%)** |
> > >
> > > ### Experiment 3: Error analysis of ChatGPT and Chameleon
> > >
> > > We added an error analysis comparing ChatGPT and Chameleon, as suggested by Reviewer yhWh. The tools in Chameleon, designed for image captioning and knowledge retrieval, notably reduce errors in these areas. Sequential tool execution further lessens errors in solution generation. GPT-4's task planning significantly outperforms ChatGPT. A detailed discussion can be found in [pdf](https://openreview.net/attachment?id=bIr8g5rvPm&name=pdf).
> > >
> > > Error categories and numbers
> > >
> > > |                     | **All** | **Tool Planning** | **Image** | **Knowledge** | **Solution** |
> > > | ------------------- | :-----: | :---------------: | :-------: | :-----------: | :----------: |
> > > | ChatGPT             |   50    |         /         |    32     |      37       |      27      |
> > > | Chameleon (ChatGPT) |   35    |        13         |    10     |       6       |      17      |
> > > | Chameleon (GPT-4)   |   28    |         1         |    19     |       3       |      8       |
> > >
> > > ### Planned revisions for our final paper
> > >
> > >
> > > In our final paper, we plan to:
> > > - **Add new experiment results and discussions**;
> > > - **Add clarifications** Chameleon's novelty and its adaptability to diverse datasets.
> > > - **Add the discussion of future work**, such as introducing APIs that allow users to define their tools and tasks, and developing skill-centered LLM agents, as suggested by Reviewer 1xpZ.
> > > - **Modify the code base** to enhance its organization.
> > > - **Add comprehensive documentation**, supplemented with demo examples in our code base, assisting users in adapting Chameleon for new tasks.
> > >
> > > We deeply value your feedback on our paper. **We sincerely hope that these new results and planned revisions heighten your appreciation of our work, prompting you to reconsider and potentially elevate your score**. This is immensely significant to our paper. Thank you!
> > >
> > > Best regards,
> > >
> > > The authors of Paper 2615

---

### Official Review · Reviewer_1xpZ · 2023-07-05

**Soundness:** 3 good
**Presentation:** 3 good
**Contribution:** 3 good
**Rating:** 7
**Confidence:** 4

**Summary:**

This work proposes Chameleon, an approach to connect large language models to different tools for compositional reasoning. Chameleon is powered by a GPT-4 planner that chains different tools to accomplish the described goal. Chameleon achieve improved performance in comparison with strong baseline models.

**Strengths:**

* The presentation of the framework is great. The paper has nice figures, clear definitions of problems, clear demonstrations of results.
* The experiments are solid. This work evaluates Chameleon on two challenging benchmarks with distinct purposes
* The study on how to make LLMs such as GPT-4 plan reasonably is useful for future research.

**Weaknesses:**

* The improvement on the two benchmarks is not as significant as described in the abstract. The abstract indicates that "on ScienceQA, improving the best published few-shot result by 11.37%" where the best-published result is a GPT-3 based approach while Chameleon is powered by GPT-4. Therefore, the comparison is not completely fair. The improvement does not necessarily come from the benefit of Chameleon. Instead, It could be as simple as GPT-4 being more powerful (as many tools are prompt-based). In Table 3, Chameleon (GPT-4) only shows a 2.55% improvement over GPT-4 CoT, and 1.62% when comparing Chameleon (GPT-3.5) with GPT-3.5 CoT.
* The paper does not explain *how* to apply Chameleon to other tasks. Even though Chameleon is claimed to be plug-and-play, there are many *task-specific* designs. For instance, the `Row_Lookup`, `Column_Lookup` , `Table_Verbalizer` tools are highly specialized for table-related questions, and they are very fine-grained. How does a practitioner know whether to have both `Row_Lookup` and `Column_Lookup`, or a simple `Table_Lookup` tool? What are the insights on tool selections? Do we need a dev set to "fine-tune" the tool pool? These details need to be further explained to make Chameleon really plug-and-play.
* Another concern is that converting the output of one tool (module) to fit the input format of another tool is pre-defined (L153-L157), indicating that it is potentially challenging to generalize Chameleon to *massive* number of tools due to the extensive human efforts.

**Questions:**

* How does Chameleon select the module inventory? What is the rule of thumb?
* How are `update_input` and `update_cache` are implemented? Any examples?
* Prompt-based tools such as `Row_Lookup` are powered by the same model as the planner; is that correct?

**Limitations:**

* Be more accurate in the improvement presentation

---

> ### Author Rebuttal · Authors · 2023-08-09
>
> Dear Reviewer,
>
> Thank you for your thoughtful analysis and encouraging words. We are pleased to know that you appreciated the great presentation of our framework. Your recognition of our solid experiments on two challenging benchmarks. Moreover, we are glad that you perceive our study on how to make LLMs plan reasonably as a useful contribution to future research.
>
> Your insights strengthen our understanding of our work's impact, and we look forward to **engaging further** with you to refine and improve our paper. Your valuable comments have been addressed as follows:
>
> ### W1: More accurate in the improvement presentation
>
> 1. **Improvement margins**: While the improvements of 2.55% and 1.62% in ScienceQA may appear modest, it's crucial to recognize that they were achieved over already highly optimized models like GPT-4 CoT (83.99%) and ChatGPT CoT (78.31%). Even minor gains can translate to significant improvements in complex tasks. Notably, for TabMWP, Chameleon improves by 9.97% and 11.25% over GPT-4 CoT and ChartGPT CoT.
>
> 2. **Comparison with GPT-3**: The intention behind our comparison with GPT-3 was to showcase Chameleon's performance using the latest models available.
>
> 3. **Clarifications including accurate presentation**: To address your concerns, we propose to: modify the wording for a more accurate performance representation; explain that comparisons are made with the latest models and justify their relevance; provide an in-depth discussion of the improvements; include experiments to compare Chameleon with GPT-3 to ensure a consistent assessment.
>
> ### W2: Apply Chameleon to other tasks
>
> 1. **Task-specific vs. plug-and-play**: We acknowledge the concern about the specialized nature of some tools. While these tools were indeed tailored for task-specific questions, Chameleon's plug-and-play design facilitates the straightforward addition of new modules that can be adapted to various tasks. To address this concern, we include a discussion illustrating its potential applicability to various domains.
>
> 2. **Tool selection insight**: Our goal here is to provide a general framework that domain experts can easily select and incorporate relevant tools and modules to significantly boost the performance of the task. They understand which tools are useful, so they select a subset of the modules that align with the tasks. We comprehend the request for more insights into tool selection. To address this concern, we will enhance our paper by including guidelines and insights on tool selection, offering a clearer path for both specialized and generalized approaches.
>
> 3. **Fine-tuning the tool pool**: The suggestion of having a development set to “fine-tune” the tool pool is valuable and aligns with our vision of making Chameleon a flexible and adaptable system. We appreciate this insight and will include discussions regarding this possibility in our revision.
>
> ### W3: Converting the output format is pre-defined
>
> We understand the concern about the potential difficulty of generalizing Chameleon to a massive number of tools. However, we would like to clarify that in Chameleon, **no special forms or extensive human effort are needed to convert the output of one tool to fit the input of another.**
>
> The outputs of each tool in Chameleon are structured entirely in *natural language* and are stored with an associated index name and value. Subsequent tools can directly utilize this textual cache if they are prompt-based or refer to the related index for other tool types. This enables *easy integration and coordination* between various tools, even as the system scales up. New tools can be added by prompting the planner with their descriptions and demonstration examples, without extensive human efforts.
>
> This thoughtful design not only distinguishes our approach from concurrent works of LLM agents but also ensures that the challenges in generalizing to a massive number of tools are mitigated.
>
> ### Q1: How does Chameleon select the module inventory?
>
> Thank you for this insightful question! Chameleon's module inventory is thoughtfully crafted to cover various reasoning abilities, such as image understanding, table comprehension, knowledge retrieval, web search, program generation, and solution generation.
>
> When adapting Chameleon for specific tasks, we assume that there are **domain experts** who want to apply LLMs to solve complex tasks, and they know certain tools are useful. So they select a subset of the modules that align with the tasks. This not only enhances planning accuracy but also optimizes computational efficiency. For example, in ScienceQA, tools related to program generation were excluded as unnecessary.
>
> ### Q2: How are update_input and update_cache implemented?
>
> They are described in L152-157, with codes detailed in the supplementary. We are pleased to elaborate further here.
>
> - `update_input`​​: ​​It is triggered by the execution of specific tools, like `Row_Lookup`, which alter or replace elements in the input to reflect the updated state.
>
> - `update_cache`: Tools such as `Image_Captioner`, `Text_Detector`, `Knowledge_Retrieval`, `Web_Search`, and `Program_Generation` generate new elements. `update_cache` stores these new elements in the cache, making them accessible for later tools' execution.
>
> ### Q3: Are prompt-based tools powered by the same model as the planner?
>
> Yes, in our current implementation, the planner and prompt-based tools utilize the same LLM base model for straightforward implementation and consistent comparison.
>
> However, Chameleon's flexibility allows them to employ different LLM base models. GPT-4 might serve as the planner for its superior planning, while ChatGPT could be employed in prompt-based tools, balancing performance and cost-effectiveness. Also, both the planner and tools can be upgraded to fine-tuned models or predefined tools for enhanced performance.

---

> ### Author Response · Authors · 2023-08-15
> **A Sincere Request for Further Feedback**
>
> Dear Reviewer,
>
> We sincerely thank you for the time and expertise you've dedicated to reviewing our paper. Your feedback has been instrumental, and we've carefully addressed each point in our rebuttal.
>
> Considering the clarifications we've provided, we kindly ask that you revisit our responses. We're hopeful that these adjustments align with your expertise and could move the paper's standing closer to *a stronger acceptance*.
>
> Your continued feedback is pivotal for us. Thank you so much!
>
> Sincerely,
>
> The authors of Paper 2615

---

> > ### Comment · Reviewer_1xpZ · 2023-08-16
> > **Reply**
> >
> > Thanks for the response, I appreciate the detailed answers. The requirement of a **domain expert** confirms my (and reviewer yhWh) concern about the "plug-and-play" feature of Chameleon. Overall, Chameleon provides a guideline for solving *specific* tasks. I would remain the same score.
> >
> > As a side note, what seems to be helpful to me is an easy-to-use software that encodes the design philosophy proposed in Chameleon so that domain experts could quickly implement pipelines for their tasks. Unfortunately, the code folder is empty in the supplementary material.

---

> > > ### Author Response · Authors · 2023-08-17
> > > **Responses to your remaining concerns**
> > >
> > > Dear Reviewer,
> > >
> > > We deeply appreciate your efforts in reviewing our paper. We highly value your new feedback and would like to address them as follows:
> > >
> > > We believe that empowering domain experts to furnish tools that assist LLMs is a cornerstone for enhancing human-AI collaboration in tackling intricate tasks. **The goal of our framework is not to build a general AI that can automatically perform every task, but to offer a practical solution that facilitates the easy adaptation of LLMs to deliver high-quality performance using existing modules**. This is especially beneficial for domain experts who might not have extensive knowledge in adapting machine learning models. This approach has not only proven practical but has also attracted wide interest, as evidenced by concurrent works listed in Table 1. The novelty and significance of our research are underscored by Reviewer yhWh, who described it as “one of the first works in the field,” and Reviewer GU39, who noted its potential to “largely help the community to explore the direction”.
> > >
> > > We wish to emphasize that **the human effort is, in fact, notably minimal compared to other alternatives that inject expert opinions**. **The only inputs expected from domain experts are**: (1) a concise natural language description outlining the task's objectives; (2) a set of tools necessary for this task; and (3) a few straightforward demonstrations of how the task employs these tools in responding to user queries. The **“plug-and-play”** feature of Chameleon facilitates the inclusion of new tools via additional descriptions and demonstrations. It also allows for the seamless updating and replacement of existing tools. Chameleon has the advantages of simplicity, adaptability, and efficiency over concurrent works. For example, Gorilla-UCB demands the assembly of tool APIs guided by human expertise, the generation of 16K instruction data, and subsequent LLM fine-tuning. HuggingGPT focuses on devising runnable demos, rather than validating the performance advantages of LLM agents, an  endeavor we undertook. Reviewer yhWh further acknowledged these merits, describing them as  **“straightforward but necessary for LLM-based agents”**.
> > >
> > > We have indeed included the code in the Supplementary Material during the paper submission. **You could access our codes via the following link**: [https://openreview.net/attachment?id=HtqnVSCj3q&name=supplementary_material](https://openreview.net/attachment?id=HtqnVSCj3q&name=supplementary_material). Our codes are well-structured, complemented by detailed instructions for execution. We have made all result files available to aid researchers in reproducing our findings and gleaning insights for upcoming research.
> > >
> > > Warm regards,
> > >
> > > The authors

---

> > > > ### Comment · Reviewer_1xpZ · 2023-08-18
> > > > **Thanks**
> > > >
> > > > Thanks to the authors for the update. I briefly walked through the code base and checked the latest response to reviewer uov202, I raised my score from 6 to 7 for the following reasons:
> > > > * Chameleon can be *relatively* general on tasks with similar components (e.g., require using a table). I appreciate the authors also added new experiments on NER and FinQA.
> > > > * The associate code base can be helpful for domain experts who have specific tasks in mind. Although currently, the code seems to be organized w.r.t datasets, I think it is a matter of time to repurpose the code to serve broader tasks. For instance, the code could provide APIs to allow users to easily define their tools and their tasks. It will be nice if the authors could provide more **design guidelines** for new tasks. For instance, how could we know whether to have both `Row_Lookup` and `Column_Lookup`, or a simple `Table_Lookup` tool?

---

> > > > > ### Author Response · Authors · 2023-08-18
> > > > > **Thanks for your acknowledgment and the raised score**
> > > > >
> > > > > Dear Reviewer,
> > > > >
> > > > > We sincerely appreciate your time spent reviewing our code and the responses to Reviewer uov2. We are encouraged by your acknowledgment of our work and your raising the score to 7.
> > > > >
> > > > > We are pleased to hear that you appreciated the general applicability of Chameleon and that our additional experiments addressed your concerns. **These discussions will be included in our revised paper**.
> > > > >
> > > > > Regarding the code base, we recognize that its current structure was organized to enhance simplicity and readability. However, as you pointed out, it allows for straightforward repurposing for broader tasks, given our modular approach. Similar components can be reused across different tasks. **We are actively modifying the code base to make it organized more generally**, as you suggested.
> > > > >
> > > > > Your idea of providing APIs to allow users to easily define their tools and tasks is very insightful. It is vital for both industry applications and research challenges. **We will discuss this idea in our revised paper**, as we believe it offers significant value for researchers and practitioners in this field. Furthermore, **we plan to delve deeper into this valuable idea in our future work** and explore how the expertise of domain experts can be naturally integrated into AI-assistant LLMs for improved human-computer interaction.
> > > > >
> > > > > Regarding your suggestion of providing design guidelines for new tasks, we completely agree. **We will offer comprehensive, hands-on documentation complete with demo examples in our code base.**  This will include step-by-step instructions along with the corresponding code implementations for developing new example tasks. We believe this documentation will assist users in adapting Chameleon for new tasks based on our existing code.
> > > > >
> > > > > To determine if a specific tool is needed, **the current strategy** is to provide all available tools with their descriptions to the system. Our qualitative study in Section 5.2 shows that LLMs like GPT-4  can deduce a new composition of tools even without a direct demonstration within the LLM prompt. **A more innovative solution** would be to categorize the tools based on specific skills. In such a setup, users would only need to identify the relevant skill category, and the system would then automatically retrieve associated tools tailored for the task at hand. For example, if the user identifies the need for the table understanding skill, the system would furnish all related tools, such as `Row_Lookup`,  `Column_Lookup`, and `Table_Lookup`. The agent itself is able to compose suitable tools for a specific query. **We will discuss the idea of skill-centered LLM agents in our revised paper and are eager to explore it in our future work**.
> > > > >
> > > > > Once again, thank you immensely for your valuable insights and continuous input on our paper!
> > > > >
> > > > > Best regards,
> > > > >
> > > > > The authors

---

### Official Review · Reviewer_yhWh · 2023-07-08

**Soundness:** 3 good
**Presentation:** 3 good
**Contribution:** 2 fair
**Rating:** 5
**Confidence:** 4

**Summary:**

The paper introduces Chameleon, a novel AI system that enhances large language models (LLMs) with plug-and-play modules for compositional reasoning. The authors argue that while LLMs have achieved remarkable progress in solving various natural language processing tasks, they have inherent limitations in accessing up-to-date information, using external tools, and performing precise mathematical and logical reasoning. To address these drawbacks, Chameleon synthesizes programs by composing various tools, such as LLMs, off-the-shelf vision models, web search engines, Python functions, and heuristic-based modules, for accomplishing complex reasoning tasks.

At the heart of Chameleon is an LLM-based planner that assembles a sequence of tools to execute to generate the final response. The authors showcase the effectiveness of Chameleon on two multi-modal knowledge-intensive reasoning tasks: ScienceQA and TabMWP. ScienceQA requires the system to answer science-related questions by integrating information from text, tables, and images, while TabMWP involves generating natural language descriptions of tables based on input queries. Chameleon, powered by GPT-4, achieves an 86.54% overall accuracy on ScienceQA, improving the best published few-shot result by 11.37%.

The authors also provide a detailed analysis of Chameleon's performance on various sub-tasks and demonstrate its ability to handle complex queries that require multiple tools and intermediate steps. They further discuss the potential of Chameleon for addressing real-world queries across various domains and improving the state of the art in multi-modal knowledge-intensive reasoning tasks. The paper concludes by highlighting the importance of enhancing current LLMs with the capability to automatically compose external tools for real-world task solving.

**Strengths:**

1. This paper probably is one of the first works in the field that explore automatic task planning and execution with GPT-4 as the agent. It further enhances the abilities of GPT-4 by leveraging various tools controlled by a GPT-4 based planner. Such a combination is straightforward but necessary for LLM-based agents.
2. One merit I like about this paper is that it has clear evaluation metrics, not like other agent-based models that merely have a runnable model and use cases. The standard evaluation clearly helps us understand the position of this paper and the performance of the proposed method.
3. The improvement on top of GPT-4/ChatgPT is significant and clearly shows the effectiveness of the proposed method. As we can see from Tables 3 and 4, the proposed method can mostly improve their direct baselines, i.e., ChatGPT and GPT-4. Also, qualitative analyses are helpful in understanding the model behavior.

**Weaknesses:**

1. Like the other recent works for LLM agents, the novelty of such kind of works is somehow ad-hoc and limited. There are many tool-augmented works, as demonstrated in Table 1, and their differences are petty nuanced, and it's hard to tell the real novelty behind them. Usually, each such kind of work would leverage the most recent LLMs, like GPT-4, to do the planning or task decomposition and call special tools to solve each subtask. Most powers come from OpenAI's model, esp. GPT-4, which is hardly claimed to be the credit of these works (but it's good to see the proposed methods can further improve GPT-4). This leaves us with an essential question, what are the core contributions of such works? Demonstrating GPT-4 can be augmented by external tools definitely is nontrivial but also engineering-heavy. For NeurIPs works, I may like to see more research values; for example, which designs of the proposed method really distinguish itself from other similar works? Is it the "generating a program in a natural-language-like format"? Or some special prompt format to trigger the planning and task execution abilities? etc etc.
2. Although the paper names its method Chameleon, trying to highlight the adaptability and versatility of LLMs, ironically, the task scope and experiment settings in this paper are limited. The planning pipeline and proposed tools are specially designed for solving the two datasets. While conceptually, the idea of augmenting GPT-4 with some tools can definitely be applied to more general tasks, this generality comes from this simple idea but not from the proposed method. I hope the authors could dedicate more effort to make the proposed method easier to be applied to more tasks.
3. Another weakness of the proposed method probably is the simple planning design, which is basically a straightforward CoT with a linear structure. As we can see from Table 3, there is still a large gap between fine-tuned STOA and the proposed method. One possible reason might be the weak planning ability of the proposed method. This is somehow related to the first weakness, meaning that most LLM agent works combine GPT-4 with tools in a very simple way. I hope the authors could explore more advanced planning algorithms to see if the performance can be further improved.

**Questions:**

I raised many questions in the weakness section, and the authors may want to resolve them and clarify some questions. The following is some minor comments:
1. I don't quite get the connection between the name "Chameleon" and the proposed method. I also don't quite get the meaning of the colors in the name.
2. I'd suggest reducing the table space, removing most of the unnecessary baselines. The only meaningful baselines probably are the STOA from previous works (mostly fine-tuned?) and GPT-4/ChatGPT.
3. Considering some improvements on top of ChatGPT and GPT-4 are kind of marginal, I'd encourage the authors to conduct a thorough error analysis to figure out how the base ChatGPT/GPT-4 make mistakes and how the proposed method reduces such errors.

---

> ### Author Rebuttal · Authors · 2023-08-09
>
> Dear Reviewer,
>
> Thank you for your insightful comments and constructive suggestions. We're grateful for your acknowledgment of our work is among the first works in the field and the distinctiveness by clear evaluation metrics. Your insights on our significant improvements and qualitative analysis are invaluable.
>
> We eagerly anticipate **further engagement** with you to strengthen our paper. Your thoughtful comments have been addressed as follows:
>
> ### W1: Novelty of the work of LLM agents
>
> We acknowledge the concern regarding the novelty in the field of LLM agents. It's important to note that many papers in Table 1 are **unpublished concurrent works**, underscoring this frontier research area. However, despite this surge in similar research, our work carves out a distinct niche due to distinct contributions:
>
> 1. **Generating natural-language-like programs**: Unlike other LLM-agent works, our system produces programs in a human-readable, natural-language format. This design mitigates errors and is more intuitive for domain experts, fostering human-machine collaboration.
>
> 2. **Versatility in API integration**: Our approach is distinctive to support a range of API types, allowing it to tackle various tasks. It sequentially executes these APIs, updating inputs and cache, without introducing complexities.
>
> 3. **Robust evaluation metrics**: We move beyond just presenting a system with “runnable examples”. Our work undergoes rigorous evaluation against real-world challenges, showing its superior performance and practical applicability.
>
> 4. **Addressing inherent LLM limitations**: Our efforts extend the LLMs’ capabilities, enabling them to navigate issues like outdated information and limitations in reasoning with non-textual data.
>
> 5. **Unpublished concurrent works**: Many papers in Table 1 represent published concurrent works, underscoring this frontier research area.
>
> *For more information, please check our clarification in the general response.*
>
> ### W2: Task scope and experiment settings
>
> Thank you for your constructive feedback on the scope and adaptability of Chameleon.
>
> 1. **Experiment settings**: The main contribution of this paper is the introduction of a framework that allows domain experts to seamlessly integrate LLMs with existing tools to enhance task performance. Therefore, under this assumption, we have validated the idea using two complex tasks. The same framework can be extended to other tasks, and additional tools can also be integrated into it.
>
> 2. **Chameleon's versatility**: The name “Chameleon” was chosen to highlight our method's adaptability. While the current implementation focuses on specific datasets, the design is meant to be applicable to a broader array of tasks. We've recorded notable performance gains in tasks not covered in the paper, such as an 11.85% gain on FinQA and an 11.39% gain on NER, both of which are benchmarks in financial reasoning.
>
> 3. **Future directions**: We value your suggestion. Our primary focus in this paper was to introduce the novel method and demonstrate its efficacy. To foster community adoption and further exploration, we will open-source the code (submitted in the supplementary), allowing other developers to utilize our framework in diverse scenarios and tasks.
>
> *For more information, please check our clarification in the general response.*
>
> ### W3: Proposed planning design and other algorithms
>
> We are grateful for your constructive feedback and appreciate the chance to clarify our design.
>
> 1. **Why the current design**: Our method was designed to be efficient and straightforward, requiring only a single LLM API call per query, avoiding feedback loops, and simplifying task performance optimization. We believe that these design decisions contribute to Chameleon's effectiveness and robustness.
>
> 2. **Comparison with existing methods**: It's worth emphasizing that Chameleon's performance is indeed commendable. It surpasses all baselines and concurrent work, including some state-of-the-art methods, which indicates that the straightforward planning design does not limit its capabilities.
>
> 3. **Potential for advanced planning algorithms**: We acknowledge your insight into the potential benefits of more complex planning algorithms. Being among the first works in the field of automatic task planning and execution with GPT-4 as the agent, we focus on the design of a general framework that allows domain experts to easily incorporate various tools. Therefore, for the sake of simplicity, we applied a straightforward planning algorithm. As the framework is general, other complex planning algorithms can be incorporated in the future.
>
> 4. **Acknowledgment of the field's challenges**: We recognize that combining GPT-4 with tools in a simple way is a common challenge in the field. Our work aims to provide a strong foundation upon which more complex strategies can be built, and your suggestions will guide our future research directions.
>
> ### Q1: Connection between the name “Chameleon” and the method
>
> Our model symbolizes its adaptability to diverse tasks and queries, adjusting its strategies much like a chameleon changes its colors. The name “Chameleon” directly relates to the title, “[Com]positional [Re]asoning with Large [Lan]guage Models”. Different colors within the name represent our model's ability to transition between various queries, aligning with the colored examples shown in Figure 1.
>
> ### Q2: Adjusting the experimental table spaces
>
> Your suggestion is highly appreciated. We will remove less performant and fine-tuned works.
>
> ### Q3: Error analysis of ChatGPT/GPT-4 and Chameleon
>
> Thank you for your valuable feedback. **We've added your suggested error analysis to the new PDF in the rebuttal period**.
>
> Chameleon's tools for image captioning and knowledge retrieval reduce mistakes in those categories. Sequential tool execution further lessens errors in solution generation. Additionally, GPT-4's task planning significantly outperforms ChatGPT.

---

> > ### Comment · Reviewer_yhWh · 2023-08-19
> > **Thanks for the rebuttal**
> >
> > I appreciate the details responses and additional experiments from the authors. My score will remain the same for the following reasons:
> > 1. The arguments from the rebuttal against my concern about the novelty don't add new and convincing information for me. Being the first batch of works in a new field doesn't mean its weakness in novelty can be neglected to some degree. The current design is more like directly applying GPT-4 to solve some datasets intuitively and I believe the authors can do better, for example, I point out that the planning and more sophisticated/general design could be potentially promising directions for improved novelty.
> > 2. The versatility of the proposed method isn't fully justified, even with the new results on "dataset-general" implementation. The whole paper is tailored in the sense that the tools, modules, and designs are specifically designed for two datasets (two other datasets are added in the rebuttal but the details are unclear and I don't think it will change the whole flow of the paper). I think this is directly against the intention behind the name Chameleon.
> > 3. I really appreciate the new error analysis from the authors and I believe it should be included in the next version. However, there is one interesting observation that GPT-4 dramatically improves the tool planning over ChatGPT, which very likely contributes to the final improvement significantly. This implies that it is GPT-4 that plays one of the most significant roles in this pipeline, which isn't surprising somehow but also leaves the question that which should be attributed to the essential success of this kind of framework, OpenAI or the authors? This question probably won't influence the acceptance of this paper (also OpenAI probably has received enough credits) but I feel should raise some certain level of concern for the community.
> > 4. Echoing reviewer uov2 somehow, I do feel the main takeaway or positioning of this paper isn't super clear to me and the overall writing oversells the contributions a little bit. When the follow-up works refer to `Chameleon`, do we expect a general framework elegantly combining GPT-4 and tools, or a variant of Auto-GPT, or a github project to solve a few specific datasets?
> >
> > Considering the divided opinions among other reviewers, regardless of whether this paper will be accepted or not, I do hope the authors can further think of the above questions (considering restructuring the paper, method, and experiments to make the work really stand out, I do like the LLM+tool idea though) and incorporate them into the revised version. Essentially, being the first batch of works doesn't mean too much, and it is the core contributions that last longer than the influence of a paper's acceptance. Thanks.

---

> > > ### Author Response · Authors · 2023-08-20
> > > **Responses and new results with Claude**
> > >
> > > We deeply appreciate your continued feedback. In response, we’d like to make clarifications with **new experiments** and hope they address your concerns.
> > >
> > > ### Novelty of our work
> > >
> > > We wish to underscore the **scientific contribution** of our work. Concurrent research studies the potential of LLMs by merely implementing runnable demos, as pointed out by you. Therefore, the extensive experiments and analysis explored in our research should not be overlooked.
> > >
> > > Advanced approaches depend on extensive computational resources and a lengthy context window, however, which were **not feasible due to the techinque limitations during our research period**. LLMs like GPT-4 were only available to a limited audience and had a maximum context window of 4K tokens.
> > >
> > > Fortunately, thanks to recent advances, LLMs have become more accessible to the broader public. The context window has expanded to 16K for GPT-4 and an impressive 100K for Claude. We are keen to delve into advanced planning designs as you've proposed. For instance, how can LLM agents refine their tool planning strategies during task execution?
> > >
> > > ### Versatility of the proposed method
> > >
> > > The versatility of our proposed method stands out when compared with concurrent works in Table 1:
> > >
> > > 1. It **supports tools of various types and skills**. Our method uses natural language to describe tools and store interaction information. In contrast, works like WebGPT depend on domain-specific programs.
> > >
> > > 2. It generates **query-specific** tool compositions, whereas works such as MM-REACT reply on  **task-specific** tool usage.
> > >
> > > 3. Its **modular design** enables both the planner and each tool to be updated or replaced individually. This differs from systems like Toolformer, which require collecting additional large-scale training data and fine-tuning the entire system. New experiments executed in a limited timeframe during the rebuttal phase further indicates its ease of development and adaptability to new datasets and settings.
> > >
> > > We will further clarify these points and reorganize the paper to provide a more coherent presentation of our method and experiments.
> > >
> > > ### Error analysis and final improvement
> > >
> > > Thank you for acknowledging our error analysis; it will be included into our revised paper.
> > >
> > > Regarding the final improvement from our proposed method, we'd like to clarify that **a fair comparison would be assessing the performance gain of Chameleon over SOTA approaches, given the same base LLM**. The base LLM primarily influences the foundational performance of various LLM approaches, but not the final improvement of our method. We've demonstrated our method's improvement on three different LLMs, with **Claude being added** during the rebuttal:
> > >
> > > The improvement using **Claude** as the base LLM:
> > >
> > > |                              | ScienceQA        | TabMWP            |
> > > | - | - | - |
> > > | Claude CoT (new)             | 78.7             | 74.0              |
> > > | Claude PoT (new)             | /                | 79.6              |
> > > | **Chameleon (Claude)** (new) | 80.8 **(+2.1%)** | 94.9 **(+15.7%)** |
> > >
> > > The improvement using **ChatGPT** as base LLM:
> > >
> > > |                         | ScienceQA (%)    | TabMWP (%)       | NER (F1)         |
> > > | - | - | - | - |
> > > | ChatGPT CoT             | 76.9             | 83.3             | /                |
> > > | ChatGPT PoT             | /                | 89.3             | 51.3             |
> > > | **Chameleon (ChatGPT)** | 78.9 **(+2.0%)** | 94.2 **(+5.1%)** | 68.1 **(+16.8)** |
> > >
> > >
> > > The improvement using **GPT-4** as base LLM:
> > >
> > > |                       | ScienceQA (%)    | TabMWP (%)       | FinQA (%)         | NER (F1)         |
> > > | - | - | - | - | - |
> > > | GPT-4 CoT             | 82.6             | 90.9             | /                 | /                |
> > > | GPT-4 PoT             | /                | 96.9             | 62.9              | 56.7             |
> > > | **Chameleon (GPT-4)** | 84.9 **(+2.3%)** | 98.8 **(+1.9%)** | 74.7 **(+11.8%)** | 66.7 **(+11.0)** |
> > >
> > > We have observed that, compared to SOTA approaches such as CoT and PoT, our Chameleon approach consistently achieves improvements across different base LLMs. For instance, there is a 1.9%-15.7% accuracy gain and an 11.0-16.8 F1 score gain. The improvement with Claude (developed by Anthropic, not OpenAI) underscores the **generality** of the Chameleon framework. This indicates that **base LLMs can be updated and replaced independently**, and, if annotation data is available, they can also be updated with fine-tuned **open-sourced** models.
> > >
> > > ### The revised version
> > >
> > > We are encouraged by your appreciation of the LLM+tool idea. We genuinely hope that our work will be impactful in the field and provide insights for future work on augmented LLM and LLM agents. To this end, we will provide refine our paper, incorporating new experiments, clarifications, and discussions. We will reorganize our code base for greater generality and provide comprehensive documentation to allow users to adapt our approach to broader tasks.

---

> ### Author Response · Authors · 2023-08-15
> **Kind request for your continued feedback**
>
> Dear Reviewer,
>
> Thank you very much for your time and efforts in reviewing our paper. We truly value your insights and have diligently addressed them in our rebuttal.
>
> As we greatly esteem your expertise, we kindly request that you take a moment to review our responses. We're eager to address any additional feedback, questions, or clarifications you might have.
>
> We're hopeful that our responses may *prompt a reconsideration of the paper's rating*. Your continued feedback is crucial to us.
>
> Sincerely,
>
> The authors of Paper 2615

---

> ### Author Response · Authors · 2023-08-19
> **Request for acknowledgment our responses and new experiments**
>
> Dear Reviewer,
>
> We have some updates relevant to your comments with three new experiments. **We hope that these new results and planned revisions will further underscore the value of our work, encouraging you to possibly adjust and elevate your rating**. This is immensely important to us. Thank you!
>
> ### Experiment 1: The performance of Chameleon in a dataset-general way
>
> We added experiments on Chameleon implemented in a dataset-general manner, as suggested by Reviewer uov2. In this setup, when assessing different datasets, both the planner and tool inventory remain consistent. The results below indicate that **Chameleon, when deployed in a dataset-general fashion, maintains similar performance gains over SOTA baselines**.
>
> New results on TabMWP:
>
> |                  | **Implementation** | **Accuracy (%)** | **Gain over SOTA** |
> | ---------------- | ---------------------- | -------------------- | ---------------------- |
> | GPT-4 CoT        | dataset-customized     | 90.9                 | /                      |
> | GPT-4 PoT        | dataset-customized     | 96.9                 | /                      |
> | Chameleon (ours) | dataset-customized     | 98.8         | +1.9%          |
> | Chameleon (ours) | **dataset-general**    | **98.5** (new)   | **+1.7%** (new)    |
>
> New results on ScienceQA:
>
> |                  | **Implementation** | **Accuracy (%)** | **Gain over SOTA** |
> | ---------------- | ---------------------- | -------------------- | ---------------------- |
> | GPT-4 CoT        | dataset-customized     | 82.6                 | /                      |
> | Chameleon (ours) | dataset-customized     | 84.9         | +2.3%          |
> | Chameleon (ours) | **dataset-general**    | **84.8** (new)   | **+2.2%** (new)    |
>
> ### Experiment 2: The performance of Chameleon on general datasets
>
> |           | **ScienceQA**  | **TabMWP**     | **FinQA (New)** | **NER (New)**   |
> | --------- | ------------------ | ------------------ | ------------------- | ------------------- |
> | GPT-4 CoT | 83.99              | 90.81              | /                   | /                   |
> | GPT-4 PoT | /                  | 96.93              | 62.87               | 56.71               |
> | Chameleon | **86.54 (+2.55%)** | **98.78 (+1.85%)** | **74.72 (+11.85%)** | **68.10 (+11.39%)** |
>
> ### Experiment 3: Error analysis of ChatGPT and Chameleon
>
> Thanks for your insightful suggestion. We have added an error analysis comparing ChatGPT and Chameleon. The tools in Chameleon, designed for image captioning and knowledge retrieval, notably reduce errors in these areas. Sequential tool execution further lessens errors in solution generation. GPT-4's task planning significantly outperforms ChatGPT. A detailed discussion can be found in [pdf](https://openreview.net/attachment?id=bIr8g5rvPm&name=pdf).
>
> Error categories and numbers:
>
> |                     | **All** | **Tool Planning** | **Image** | **Knowledge** | **Solution** |
> | ------------------- | :-----: | :---------------: | :-------: | :-----------: | :----------: |
> | ChatGPT             |   50    |         /         |    32     |      37       |      27      |
> | Chameleon (ChatGPT) |   35    |        13         |    10     |       6       |      17      |
> | Chameleon (GPT-4)   |   **28**    |         **1**         |    **19**     |       **3**       |      **8**       |
>
> ### Planned revisions for our final paper
>
>
> In our final paper, we plan to:
> - **Add new experiment results and discussions**;
> - **Add clarifications** Chameleon's novelty and its adaptability to diverse datasets.
> - **Add the discussion of future work**, such as introducing APIs that allow users to define their tools and tasks, and developing skill-centered LLM agents, as suggested by Reviewer 1xpZ.
> - **Modify the code base** to enhance its organization.
> - **Add comprehensive documentation**, supplemented with demo examples in our code base, assisting users in adapting Chameleon for new tasks.
>
> Best regards,
>
> The authors of Paper 2615

---

### Author Rebuttal · Authors · 2023-08-09

We appreciate the efforts of all reviewers in reviewing our paper. Thank you for the insightful comments that recognize the novelty and contributions of our work, and for the valuable suggestions to enhance our paper. We would like to address the common concerns, emphasize the novelty and contribution of our paper, and the rules for benchmark selection as follows:

### Common concerns

Some critical concerns were shared by reviewers, including: 1) the general framework, 2) how easy it is to adapt to others and why, 3) why the two chosen datasets are useful to support our general argument, and 4) the reasoning for a dataset-specific module inventory at this time.

Regarding these concerns, both Reviewer yhWh (R1) and GU39 (R3) have provided comments regarding the substantial novelty in research present in our work. We are grateful to Reviewer yhWh (R1) and 1xpZ (R2)  for recognizing our research as a well-conducted technical study that presents comprehensive and robust results. Likewise, we appreciate R3's acknowledgment of the excellence of our model design.

We would like to emphasize that while expanding Chameleon to more datasets may seem like a relatively apparent next step, we are the first to come up with a concrete design and demonstrate its effectiveness. Publishing these results will put everyone else in the field one step ahead, and we believe that these findings are valuable for advancing this field.

### Novelty of our approach

Our Chameleon work represents a significant stride in the evolving landscape of LLMs, as acknowledged by the reviewers in various aspects of the novelty:

1. **One of the first works in the field of LLM agents**: As Reviewer yhWh (R1) recognized, our approach is pioneering in automatic task planning and execution using GPT-4 as the agent. The combination of GPT-4 with various controlled tools enhances the capabilities of existing LLMs, marking a significant advancement in this domain.

2.  **An innovative approach that addresses the inherent limitations**: Reviewer GU39 (R3) acknowledged our novel system, Chameleon, as a unique solution that addresses fundamental challenges faced by LLMs.

3. **A general-purpose framework of compositional reasoning**: The Chameleon framework is not confined to specific tasks but rather has the potential to be extended as a general application, as pointed out in the takeaway by Reviewer uov2 (R4).

4. **Modularity with natural-language-format program generation**: Reviewer uov2 (R4) also highlighted the distinction of our work in constructing a solution entirely within modules, indicating the advantages of performing more tightly controlled reasoning steps and introducing constraints on the sequence of modules to use.

### Scientific contribution of our approach

Our scientific contributions are summarized as follows:

1. **Exploration of a critical research gap**: Reviewer uov2 (R4) validates the significance of this exploration: “The research topic (how to best use LLMs with tools) is definitely significant.” Our research fills a notable gap in the literature, particularly regarding comprehensive evaluations of LLMs with tools. Costly baselines have been evaluated and compared with published results, ensuring robustness and credibility.

2. **Rigorous evaluations**: Our work stands out for its clear and standardized evaluation metrics, differentiating it from other agent-based models that may lack clear benchmarks, as recognized by Reviewer yhWh (R1).

3. **Demonstrable enhancements and benchmarking:**: The study has consistently demonstrated significant improvements over strong baseline models, as acknowledged by multiple reviewers.

4. **Detailed experimental and qualitative insights**: The paper's strength also lies in its comprehensive experimental approach and the qualitative analysis that provides deeper insights into related research, as acknowledged by all reviewers.

### Concerns regarding dataset selection

Our study's emphasis on **compositional reasoning and tool use** guided the careful selection of benchmarks. The following factors were key considerations in choosing ScienceQA and TabMWP as benchmarks:

1. ​​**Multimodal contexts and reasoning abilities**: Both ScienceQA and TabMWP involve multimodal contexts and require the use of varied tools for reasoning abilities. Unlike other popular datasets like GSM8K, which focus on text-only problems, these benchmarks assess more complex capabilities. They align closely with the core research goals of our study, examining LLM's abilities in compositional reasoning and tool use, rather than merely text processing or domain-specific tasks.

2. **Diversity of questions, domains, contexts, and skills**: They feature a rich and varied composition of tools, aligning with our focus on compositional reasoning. Other datasets that are domain or context-limited do not provide the same level of diversity and challenge.

3. **Far from saturation in performance evaluation**: These benchmarks are not yet saturated in terms of performance, as evidenced by the latest few-shot models' scores (83.99% in ScienceQA for GPT-4 CoT and 81.8% in TabMWP for Codex PoT-SC). Other datasets have reached saturation, such as ARC, HellaSwag, and GSM-8K, with GPT-4's high scores of 96.3%, 95.3%, and 92.0%, respectively.

4. **Recent introduction and reduced risk of contamination**: Proposed in late 2022, these benchmarks carry less risk of contamination to the latest LLM training, ensuring a more genuine evaluation of the models' capabilities.

5. **Clear evaluation metrics**: Their focus on multi-choice or numeric questions allows for clear, objective evaluation metrics, aligning with our commitment to rigorous assessment.

###  Error analysis of ChatGPT and Chameleon

As suggested by Reviewer yhWh (R1), we added an error analysis of ChatGPT and Chameleon, to the new PDF below in the rebuttal period.

---

### Author Response · Authors · 2023-08-12
**Kind Request for Discussions and Feedback for Paper 2615**

Dear Reviewers,

Firstly, we'd like to express our sincere gratitude for taking the time to review our paper and for providing invaluable feedback. We have made our best efforts to address all the comments raised during the initial review.

If you've had a chance to go through our responses, we kindly request your feedback. If our responses have satisfactorily addressed your concerns, we hope you might consider adjusting your ratings to reflect these changes.

Your engagement during this discussion period is crucial for the improvement of our work. We genuinely value your insights and look forward to your continued feedback.

Thank you for your time and consideration.

Warm regards,

Authors of Paper 2615

---

### Decision · Program_Chairs · 2023-09-21

**Decision:**

Accept (poster)

**Comment:**

The paper proposes a methodology for leveraging LLMs to automatically compose and combine results from a range of tools.  While the technical contributions are relatively straightforward, the topic is highly relevant to current LLM research, and the overall framework represents a substantial engineering effort that achieves state-of-the-art results on ScienceQA and TabMWP with additional results for two datasets reported in the rebuttal (among other extensive experimental analysis that has been further augmented on rebuttal to address reviewer concerns).  I found the failure analysis in supplementary material to be very helpful and suggestive of follow-on research directions

There were a number of concerns that were raised in reviews and discussed on rebuttal, mostly to the reviewers' satisfaction.  I think two concerns are worth mentioning here.  One major concern involves overlap with the HuggingGPT paper posted to Arxiv on March 30, 2023, which would be considered concurrent work according to the 2023 NeurIPS FAQ for Authors.  A second major concern for reviewers is that some aspects of Chameleon are dataset-specific (tool availability, ordering), which the authors claim to address in new experiments during the rebuttals, but which do represent a major change from the submitted version.

While the paper is recommended for acceptance, reviewers maintain concerns about significant revisions to the paper to incorporate rebuttal content and the authors are requested to carefully consider such changes at the final revision stage that will not undergo further review.